# ELASTIC MoE: UNLOCKING THE INFERENCE-TIME SCALABILITY OF MIXTURE-OF-EXPERTS

## ABSTRACT

Mixture-of-Experts (MoE) models typically fix the number of activated experts $k$ at both training and inference. Intuitively, activating more experts at inference $k'$ (where $k' > k$) means engaging a larger set of model parameters for the computation and thus is expected to improve performance. However, contrary to this intuition, we find the scaling range to be so narrow that performance begins to degrade rapidly after only a slight increase in the number of experts. Further investigation reveals that this degradation stems from a lack of learned collaboration among experts. To address this, we introduce Elastic Mixture-of-Experts (EMoE), a novel training framework that enables MoE models to scale the number of activated experts at inference without incurring additional training overhead. By simultaneously training experts to collaborate in diverse combinations and encouraging the router for high-quality selections, EMoE ensures robust performance across computational budgets at inference. We conduct extensive experiments on various MoE settings. Our results show that EMoE significantly expands the effective performance-scaling range, extending it to as much as 2-3$\times$ the training-time $k$, while also pushing the model's peak performance to a higher level.

## 1 INTRODUCTION

Large-scale models based on the Transformer architecture (Vaswani et al., 2017) have demonstrated remarkable performance across a wide range of tasks (OpenAI, 2023; Touvron et al., 2023b;a). However, this performance gain is often accompanied by a substantial increase in model size, leading to prohibitive computational costs for both training and inference. To address this challenge, the Mixture-of-Experts (MoE) paradigm (Fedus et al., 2022; Lepikhin et al., 2021) has garnered significant attention. By employing a sparsely activated architecture, MoE models effectively maintain model capacity while enhancing computational efficiency, leading to its widespread adoption.

Most MoE models (DeepSeek-AI et al., 2024; Team, 2024; Dai et al., 2024; Team et al., 2025) are typically implemented via a Top-$k$ strategy (Shazeer et al., 2017), where a fixed number of experts $k$ is selected for each token. This design ensures a predictable computational budget for training and keeping this number fixed for inference. This prompts a natural question: if a larger computational budget is available at inference, could performance be enhanced by activating more experts? Intuitively, leveraging a larger number of experts at inference time means engaging a larger set of model parameters for computation and thus is expected to improve performance.

We uncover an intriguing and previously under-explored phenomenon: when a model is trained with $k$ experts, the effective scaling range at inference is so narrow that increasing this to a slightly larger $k'$ ($> k$) causes performance to degrade rapidly. While increasing the number of activated experts during training offers some partial relief, we observe that such models collapse once inference returns to the original $k$ budget. Upon further analysis, we identify the root cause of this inability to extrapolate to larger $k'$ values as disparities in expert co-occurrence frequencies. Specifically, the additionally activated experts at inference have not been trained to collaborate effectively with the originally selected experts, as these new combinations are rarely encountered during training. This lack of learned collaboration causes the observed performance degradation.

In this paper, we introduce Elastic Mixture-of-Experts (EMoE), a novel training framework that equips MoE models with computational flexibility. The effectiveness of EMoE stems from two key designs. First, to address co-activation failure, we propose *stochastic co-activation sampling*, which

draws inspiration from Monte Carlo sampling to stochastically select diverse expert combinations during training. This strategy efficiently increases the co-occurrence frequency of expert combinations without incurring significant training overhead, thereby enabling the model to learn collaborative capabilities required for effective inference with high expert counts. Second, to ensure reliable performance across varying computational budgets, we introduce the *hierarchical router loss*. This loss leverages KL divergence to push the router's output distribution away from uniformity, thereby imposing a clear hierarchical ranking upon the experts for each token. This yields a high-quality set of top-$k$ experts across budgets, allowing the model to scale gracefully with available computation.

We conduct extensive experiments to validate the performance and adaptability of our EMoE framework across LoRA-based and FFN-based MoE scenarios. These experiments are performed on three model architectures with varying parameter scales, and performance is assessed across nine benchmark datasets. Results show that, unlike standard Top-$k$ models, EMoE achieves significant gains across a much wider range of activated expert counts during inference, which can reach up to 2-3$\times$ the training-time $k$. Moreover, it consistently outperforms baselines under various computational budgets ($k'$), highlighting its strong utility in diverse settings. Further experimental analysis confirms that both stochastic co-activation sampling and the hierarchical router loss are crucial to EMoE's effectiveness. In summary, these results collectively establish EMoE as a powerful and practical framework that successfully unlocks the elastic potential of MoE models during inference.

## 2 RELATED WORK

**Mixture-of-Experts**   The Mixture-of-Experts (MoE) architecture is an efficient model design that increases model capacity while controlling computational costs by activating only a subset of parameters for each input. The idea is first introduced by Jacobs et al. (1991) and later popularized in deep learning through the work of Shazeer et al. (2017). Subsequent developments such as GShard (Lepikhin et al., 2021) and Switch Transformer (Fedus et al., 2022) demonstrate that replacing the feed-forward layers in Transformers with MoE layers enables efficient pre-training at the trillion-parameter scale, achieving remarkable results. Most follow-up research has since focused on key areas like optimizing expert design (DeepSeek-AI et al., 2024; Wang et al., 2024a), routing mechanisms (Puigcerver et al., 2024; Wang et al., 2025), and load balancing strategies (Wang et al., 2024b). Recently, several studies (Huang et al., 2024; Zeng et al., 2024; Jin et al., 2025) explore dynamic routing, where the number of activated experts varies per token to allocate more computation to complex tokens and less to simpler ones, all under a fixed computational budget. Different from previous studies, our work does not focus on redistributing computation under a fixed budget. Instead, we explore how to ensure and enhance the performance of MoE models when the total computational budget changes during inference. Our goal is to endow MoE models with inference-time scalability.

**Inference-Time Computational Scaling**   Scaling computation at inference is a strong strategy for addressing the performance-efficiency trade-off in LLMs (Snell et al., 2024). Current studies mainly explore two dimensions: the depth dimension, which aims to enhance model ability by increasing the length of the reasoning chain during inference (Chen et al., 2025; Bae et al., 2025); and the width dimension, where prior work primarily on dense models extracts subnetworks of varying sizes from a pre-trained large model by drawing on the concept of pruning (Devvrit et al., 2024; Haberer et al., 2024), to accommodate different hardware or latency constraints. Our work adopts a fundamentally new perspective on inference-time scalability for MoE models. Instead of increasing model depth or extracting sub-models, we are the first to explore how to effectively utilize increased computational budgets by activating and combining a greater number of experts. This compositional approach to computation scaling at inference allows the model to transition smoothly from a sparse activation state to a denser one, thereby unlocking its full potential in accordance with available resources.

## 3 AN EMPIRICAL STUDY ON INFERENCE-TIME EXPERT SCALING

### 3.1 PRELIMINARIES

A standard MoE layer is composed of two primary components: a set of $N$ independent expert networks $\{E_i(\cdot;\theta_i)\}_{i=1}^N$, and a router network $G$, which dynamically selects a sparse combination of these experts for each input token. Each expert $E_i$ is a neural network (e.g., an FFN or a LoRA

module (Hu et al., 2022)) parameterized by $\theta_i$. Given an input token representation $x$, the router produces logits $h(x)$ that determine the assignment of the token to the experts. This is typically done via a linear mapping $h(x) = \mathbf{W}_g x$, where $\mathbf{W}_g$ is the router's learnable matrix. To enforce sparsity and reduce computation, the standard approach is Top-$k$ gating, where a fixed number $k$ of experts are selected for each token. Let $\pi(x)$ denote the permutation that sorts the logits $h(x)$ in descending order. The set of Top-$k$ active experts $\mathcal{S}_k(x)$ is then defined as:

$$\mathcal{S}_k(x) = \{\pi_1(x), \pi_2(x), \ldots, \pi_k(x)\}. \tag{1}$$

The final output of the MoE layer is a weighted combination of the outputs from these active experts. The weights are derived from the router's logits, which are normalized via a softmax function applied only over the selected experts. The final output $y(x)$ is formulated as:

$$y(x) = \sum_{i \in \mathcal{S}_k(x)} \frac{\exp(h_i(x))}{\sum_{j \in \mathcal{S}_k(x)} \exp(h_j(x))} \cdot E_i(x; \theta_i). \tag{2}$$

### 3.2 FINDINGS AND ANALYSIS

The static Top-$k$ design of standard MoE models raises a foundational question regarding their flexibility. Given additional computational resources at inference, a natural strategy for performance enhancement would be to activate more experts, thereby leveraging a larger portion of the model's total capacity. Intuitively, this should improve model performance. To investigate this, we train a LLaMA2-7B (Touvron et al., 2023b) model equipped with LoRAMoE (Dou et al., 2023) containing 32 experts. We train separate models, each with a different number of activated experts $k$.

As shown in Figure 1, our initial experiments reveal a counter-intuitive phenomenon: when the number of experts activated at inference $k'$ exceeds the training-time budget (e.g., $k = 2$), performance holds briefly but quickly drops thereafter, even though more parameters are being utilized. Using a larger training budget (e.g., $k = 4$), the performance peak shifts to a higher budget and the decline occurs later on the $k'$ axis. However, overly large $k$ configurations incur prohibitive computational overhead, and pushing $k$ further introduces another failure mode: such models perform poorly when the inference budget is reduced ($k' < k$), leading to performance collapse when returning to the original inference budget. Empirically, this occurs because the model learns to rely solely on activating many experts simultaneously, without teaching the router how to make effective selections under conventional budgets.

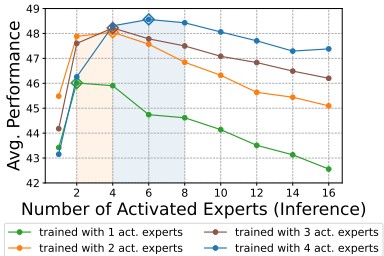

Figure 1: Performance of MoE models trained with fixed $k$ under varying inference-time activated experts ($k'$). The color regions show where optimal performance briefly holds.

These findings motivate our central goal: to develop a method that provides flexibility across different inference-time budgets while preserving the standard training cost of conventional $k$ configurations.

To diagnose the cause of the inability of models trained with $k$ to extrapolate to larger $k'$, we investigate the discrepancy in expert activation patterns between training and inference. We introduce an expert co-occurrence matrix, $M^{(k)} \in \mathbb{R}^{N \times N}$, to quantify the frequency with which any two experts are activated together for the same token. The matrix is defined as:

$$M_{ij}^{(k)} = \frac{1}{|D|} \sum_{x \in D} \mathbf{1}[i \in \mathcal{S}_k(x) \wedge j \in \mathcal{S}_k(x)], \tag{3}$$

where $D$ is the dataset, and $\mathcal{S}_k(x)$ is the set of experts selected by Top-$k$ gating for a given token $x$. Figures 2a visualize these co-occurrence matrices for models trained with $k = 2$ and $k = 6$ experts, respectively, across different layers. A significant disparity emerges. For the model trained with $k = 2$, the co-occurrence matrix observed during training is sparse, reflecting a specific set of learned expert pairings. However, when this model is subjected to inference with $k' = 6$, the matrix becomes much denser and qualitatively different. This indicates that the model is forced to utilize many expert combinations that are seldom, if ever, seen during training. These experts have not been optimized to collaborate, leading to a breakdown in their collective output. Conversely, for the model trained with $k = 6$ (Figures 2b), the co-occurrence matrices from training and inference are structurally similar.

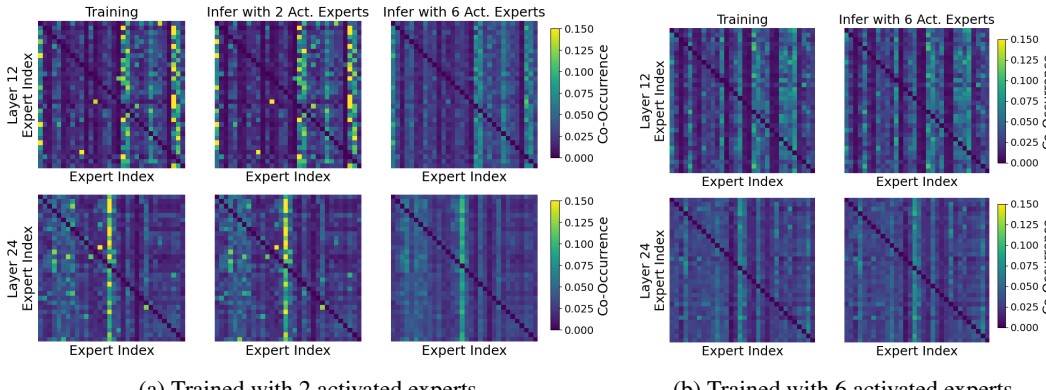

(a) Trained with 2 activated experts                (b) Trained with 6 activated experts

Figure 2: Visualization of expert co-occurrence matrices. Panels show models trained with (a) $k = 2$ and (b) $k = 6$ experts. Each panel compares the co-occurrence patterns observed during training and inference. Extrapolating from $k = 2$ to $k' = 6$ substantially changes the co-activation structure.

This alignment between training and inference conditions explains why the model's performance is more stable. We therefore hypothesize that the inability of MoE models to extrapolate to higher expert counts stems from a lack of collaborative training among the sparsely activated experts.

To quantify the impact of this co-occurrence disparity, we measure the Frobenius norm of the distance between the co-occurrence matrix from training, $M^{(k)}$, and the one from inference, $M^{(k')}$:

$$\Delta(k \rightarrow k') = \|M^{(k)} - M^{(k')}\|_F. \quad (4)$$

This metric captures the distance in expert activation patterns. A small $\Delta$ indicates that the expert combinations encountered at inference are similar to the distribution seen during training. Conversely, a large $\Delta$ signifies a severe distributional shift, where the model is used on untested expert collaborations. Figure 3 plots this relationship for a model trained with $k = 2$ experts. The results show a clear and compelling trend. As the number of activated experts at inference ($k'$) increases, the F-norm distance $\Delta$ grows monotonically, which is anti-correlated with model performance. Beyond the optimal point, every subsequent increase in the number of experts leads to a larger co-occurrence distance and a corresponding, significant drop in performance. This means that one of the main reasons for the performance degradation observed when extrapolating to more experts is that using new expert combinations that are not sufficiently trained to handle.

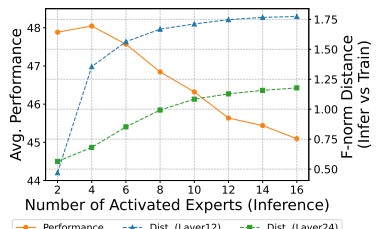

Figure 3: Co-occurrence distance vs. model performance for a model trained with $k = 2$.

## 4 ELASTIC MIXTURE-OF-EXPERTS

To unlock inference-time scalability without incurring the prohibitive training overhead, we introduce Elastic Mixture-of-Experts (EMoE) as shown in Figure 4. EMoE incorporates two designs: stochastic co-activation sampling, which resolves the expert co-occurrence discrepancy by training diverse combinations of experts to collaborate effectively, and a hierarchical router loss, which regularizes the router to produce stable and decisive expert rankings for each token. Together, these designs ensure robust performance across varying computational budgets.

### 4.1 STOCHASTIC CO-ACTIVATION SAMPLING

Our pilot study reveals that the inability of models trained with $k$ to extrapolate to larger $k'$ stems from insufficient collaborative training among experts, with certain expert combinations rarely encountered during training with a $k$. The ideal solution would be to train with a large number of experts $k_{\text{ideal}}$ for

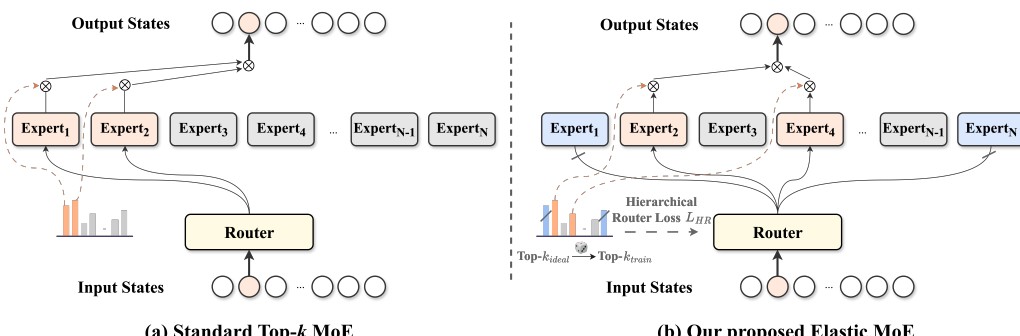

Figure 4: Comparison of the standard Top-$k$ MoE and our Elastic Mixture-of-Experts (EMoE). EMoE is designed to unlock scalability at inference time. For each input, it first forms a candidate pool $\mathcal{S}_{k_{\text{ideal}}}$ of top-scoring experts. A smaller subset $\mathcal{S}_{\text{co-act}}$ is then uniformly drawn from this pool for computation. The total objective combines standard MoE losses with the hierarchical router loss $\mathcal{L}_{\text{HR}}$, which regularizes the router to produce a decisive, non-uniform expert distribution.

every token. Given input $x$, the MoE output would be:

$$y_{\text{ideal}}(x) = \sum_{i \in \mathcal{S}_{k_{\text{ideal}}}(x)} \frac{\exp(h_i(x))}{\sum_{j \in \mathcal{S}_{k_{\text{ideal}}}(x)} \exp(h_j(x))} \cdot E_i(x; \theta_i). \tag{5}$$

However, implementing this is computationally prohibitive, as it defeats the purpose of a sparse MoE. To approximate this co-activation training without the associated cost, we therefore introduce stochastic co-activation sampling. Inspired by Monte Carlo sampling, it approximates the ideal training objective by sampling a small subset of experts from a larger candidate pool of experts $\mathcal{S}_{k_{\text{ideal}}}(x)$. Specifically, for each token $x$, we draw a subset of size $k_{\text{train}} < k_{\text{ideal}}$:

$$\mathcal{S}_{\text{co-act}}(x) \sim \text{UniformSample}(\mathcal{S}_{k_{\text{ideal}}}(x), k_{\text{train}}), \tag{6}$$

and compute the MoE output $y_{\text{co-act}}(x)$ on this subset. Co-activation sampling provides a stochastic approximation to the ideal objective over multiple training steps, effectively capturing diverse expert co-activation patterns without the prohibitive cost of large-$k$ training.

To further ease the optimization burden, we introduce a dynamic sampling process in practice. This strategy replaces the fixed-size $k_{\text{ideal}}$ with a variable one, adjusting the sampling space to stabilize training. For each input token, we stochastically determine the size of a candidate pool $\tilde{k}_{\text{ideal}}$, by drawing it uniformly from the integer interval $[k_{\text{train}}, k_{\text{ideal}}]$. It ensures that the candidate pool is frequently drawn from a smaller, higher-confidence set (i.e., when $\tilde{k}_{\text{ideal}}$ is sampled to be close to $k_{\text{train}}$) and guarantees that the core group of top experts receives consistent and focused training signals. Concurrently, the uniform sampling up to $k_{\text{ideal}}$ introduces controlled exploration, allowing the model to learn diverse co-activation patterns. From this dynamically sized candidate pool, we then perform the above sampling step in Eq 6, selecting a final training subset $\mathcal{S}_{\text{co-act}}(x)$.

**Why Co-activation Sampling Works?** The efficacy can be directly understood by examining its impact on the expert co-occurrence matrix $M^{(k)}$ from our pilot study. We previously established that performance degradation when scaling from a training budget $k$ to an inference budget $k'$ correlates strongly with a large co-occurrence distance $\Delta(k \to k') = \|M^{(k)} - M^{(k')}\|_F$. This distance arises because many entries in the matrix $M_{ij}^{(k)}$ during training are zero or near-zero, while the corresponding entries $M_{ij}^{(k')}$ become substantially non-zero at inference, forcing the model to rely on untested expert combinations.

Our proposed co-activation sampling is designed to minimize this future discrepancy by "filling in" the sparse training co-occurrence matrix in advance. The mechanism is probabilistic. For any token where two experts $i$ and $j$ fall within the candidate pool $\mathcal{S}_{k_{\text{ideal}}}(x)$, their probability of being jointly selected for a training update is uniformly defined as:

$$P(i, j \in \mathcal{S}_{\text{co-act}}(x) \mid i, j \in \mathcal{S}_{k_{\text{ideal}}}(x)) = \frac{C(k_{\text{ideal}} - 2, k_{\text{train}} - 2)}{C(k_{\text{ideal}}, k_{\text{train}})}. \tag{7}$$

This ensures that a wide range of expert pairs receive collaborative training signals. Let's revisit our concrete example ($N = 32$ experts, standard training $k = 2$, versus EMoE with $k_{\text{train}} = 2, k_{\text{ideal}} = 8$). Consider an expert pair $(i, j)$ where one or both experts are ranked outside the top-2 but within the top-8. In standard training, their co-occurrence entry $M_{ij}^{(2)}$ is effectively zero. With co-activation sampling, assuming this pair is in the top-8 pool for a given token, their co-activation probability becomes $C(6, 0)/C(8, 2) = 1/28 \approx 3.6\%$. While this probability seems small for a single instance, when aggregated over multiple training steps, it guarantees that the corresponding entry $(M_{\text{co-act}}^{(2)})_{ij}$ becomes substantially non-zero, mitigating the distance with the co-occurrence matrix at inference.

### 4.2 HIERARCHICAL ROUTER LOSS

Recall that the router does not reliably make effective selections under low-activation budgets, leading to degraded performance (Figure 1), and the inference budget $k'$ can vary in practical deployments, we expect the model to achieve strong performance across different budget levels. A key issue arises when the router assigns nearly uniform weights to experts: in this case, the distinction between Top-$k$ and the rest becomes ambiguous, and a small $k'$ will underperform. Therefore, the router should produce a clear, hierarchical expert ranking for each token, such that activating only a few experts (small $k'$) or many experts (large $k'$) both lead to reliable performance. To achieve this, we encourage the router distribution $h(x)$ to be far from a uniform distribution. Concretely, we introduce a KL-based regularization:

$$\mathcal{L}_{\text{HR}} = -D_{KL}(h(x) \,\|\, \mathcal{U}) = -\sum_{i=1}^{N} h_i(x) \log\left(\frac{h_i(x)}{1/N}\right), \tag{8}$$

where $\mathcal{U}$ denotes the uniform distribution over experts. Here we use reverse KL rather than forward KL. Using forward KL, i.e., $-D_{KL}(\mathcal{U} \,\|\, h(x)) = \frac{1}{N}\sum_i(\log h_i(x) + \log N)$, the resulting gradients would be $-\frac{\partial}{\partial h_i}D_{KL}(\mathcal{U} \,\|\, h(x)) = \frac{1}{N h_i(x)}$. In contrast, reverse KL yields smoother gradients $\partial \mathcal{L}_{\text{HR}}/\partial h_i = -\log(h_i(x)N) - 1$. The gradients of forward KL increase more rapidly as $h_i(x)$ approaches zero, since $1/h_i(x)$ diverges much faster than $-\log h_i(x)$ does. For example, when $h_i(x) = 0.01$, we have $1/h_i(x) = 100$, while $-\log h_i(x) \approx 4.6$. This sharp increase in gradients can cause instability during training. By contrast, the reverse KL sharpens the distribution without excessive concentration, preserving stable Top-$k$ rankings while maintaining the potential contribution of other experts. Thus, the full EMoE training objective integrates the cross-entropy loss $\mathcal{L}_{ce}$, load balance loss $\mathcal{L}_b$, and our proposed loss $\mathcal{L}_{\text{HR}}$: $\mathcal{L} = \mathcal{L}_{ce} + \mathcal{L}_b + \lambda \cdot \mathcal{L}_{\text{HR}}$, where $\lambda$ is a coefficient balancing the objectives.

**Putting them together.** EMoE provides a lightweight yet powerful framework for training inference-scalable MoE models. Stochastic co-activation sampling directly tackles the problem of co-activation failure by teaching experts to collaborate within diverse, stochastically sampled combinations. Concurrently, the hierarchical loss guides the router to learn a stable and decisive expert ranking. Together, they ensure that the model can gracefully and effectively scale its performance to match the given computational budget at inference time, eliminating the need to train or deploy multiple MoE variants tailored to different computational settings. Notably, EMoE maintains the same training cost $k_{\text{train}}$ as the Top-$k$ method. A detailed analysis of training cost is provided in Appendix A.2.

## 5 EXPERIMENTS

### 5.1 EXPERIMENTAL SETUP

**Model Settings.** Our experiments consider two common MoE scenarios: LoRA-based and FFN-based settings. In the LoRA-based scenario, we adopt LLaMA2-7B (Touvron et al., 2023b) as the base model and configure 32 LoRA experts in each layer. In the FFN-based scenario, we evaluate three advanced MoE models of different scales: OLMoE-0924 (Muennighoff et al., 2024), DeepSeekV2-Lite (DeepSeek-AI et al., 2024), and ERNIE-4.5-21B-A3B (Team, 2025). Their parameter sizes are 7B-A1B, 16B-A2.4B, and 21B-A3B, respectively.

**Baselines.** We compare against two categories of baselines. The first category consists of mainstream MoE models that employ a fixed Top-$k$ strategy. The second category includes dynamic

Table 1: Comparison between EMoE and the Top-$k$ method across different numbers of activated experts ($k'$) during inference. For each base model, both methods are trained with the same budget. All experiments are repeated three times, and we report the mean results along with the standard deviation.

| | ARC-c | ARC-e | GSM8K | HellaS. | HumanE. | NQ | Tri.QA | Wino. | MMLU | AVG |
|---|---|---|---|---|---|---|---|---|---|---|
| *LoRAMoE (trained with 2 activated experts)* | | | | | | | | | | |
| Top-$k$ ($k'=1$) | 56.95 | 73.55 | 33.33 | 52.71 | 15.97 | 22.89 | 54.32 | 53.59 | 46.04 | $45.48_{\pm0.91}$ |
| Top-$k$ ($k'=2$) | 56.54 | 75.98 | 37.54 | 54.87 | 20.24 | 25.55 | 57.62 | 55.06 | 47.55 | $47.88_{\pm0.57}$ |
| Top-$k$ ($k'=4$) | 56.75 | 76.65 | 38.16 | 53.07 | 19.39 | 26.91 | 59.06 | 55.19 | 47.23 | $48.05_{\pm0.36}$ |
| Top-$k$ ($k'=6$) | 57.74 | 75.60 | 35.94 | 50.76 | 18.70 | 27.17 | 59.53 | 55.49 | 47.20 | $47.57_{\pm0.52}$ |
| EMoE ($k'=1$) | 55.25 | 71.78 | 31.24 | 51.52 | 17.68 | 25.46 | 54.79 | 56.75 | 46.64 | $45.68_{\pm1.19}$ |
| EMoE ($k'=2$) | 56.95 | 78.66 | 37.98 | 53.57 | 18.90 | 26.62 | 58.20 | 55.41 | 47.65 | $48.22_{\pm0.62}$ |
| EMoE ($k'=4$) | 58.31 | 79.72 | 38.21 | 55.95 | 18.29 | 27.78 | 59.24 | 55.64 | 47.89 | $49.00_{\pm0.70}$ |
| EMoE ($k'=6$) | 60.34 | 79.37 | 38.21 | 56.14 | 20.73 | 27.87 | 59.77 | 55.33 | 47.73 | $49.50_{\pm0.65}$ |
| *OLMoE-1B-7B-0924 (trained with 8 activated experts)* | | | | | | | | | | |
| Top-$k$ ($k'=4$) | 43.73 | 65.96 | 17.13 | 41.40 | 13.41 | 14.57 | 32.04 | 50.36 | 39.17 | $35.31_{\pm1.06}$ |
| Top-$k$ ($k'=8$) | 52.54 | 71.78 | 24.94 | 48.68 | 21.34 | 17.95 | 38.98 | 52.09 | 43.85 | $41.35_{\pm0.55}$ |
| Top-$k$ ($k'=16$) | 53.56 | 72.31 | 25.85 | 48.59 | 17.07 | 18.12 | 39.10 | 51.62 | 43.61 | $41.09_{\pm0.75}$ |
| EMoE ($k'=4$) | 45.08 | 72.84 | 21.83 | 42.82 | 15.24 | 14.32 | 34.43 | 51.93 | 37.78 | $37.36_{\pm1.29}$ |
| EMoE ($k'=8$) | 51.86 | 75.49 | 26.54 | 50.42 | 20.12 | 19.34 | 40.99 | 53.43 | 40.29 | $42.05_{\pm0.67}$ |
| EMoE ($k'=16$) | 55.93 | 74.25 | 27.98 | 51.84 | 18.90 | 18.31 | 41.61 | 52.64 | 41.80 | $42.58_{\pm0.92}$ |
| *DeepSeek-V2-Lite (trained with 6+2 activated experts)* | | | | | | | | | | |
| Top-$k$ ($k'=3+2$) | 58.98 | 72.66 | 42.99 | 57.36 | 28.66 | 18.89 | 41.36 | 55.96 | 47.76 | $47.18_{\pm0.48}$ |
| Top-$k$ ($k'=6+2$) | 64.07 | 75.49 | 49.36 | 58.52 | 36.59 | 21.63 | 46.86 | 57.22 | 49.88 | $51.07_{\pm0.01}$ |
| Top-$k$ ($k'=12+2$) | 62.71 | 74.43 | 50.19 | 57.60 | 39.02 | 21.05 | 46.38 | 57.22 | 49.89 | $50.94_{\pm0.16}$ |
| EMoE ($k'=3+2$) | 58.64 | 79.37 | 47.01 | 60.06 | 34.15 | 22.16 | 47.31 | 57.46 | 48.33 | $50.50_{\pm0.69}$ |
| EMoE ($k'=6+2$) | 62.71 | 82.72 | 47.92 | 62.48 | 40.24 | 23.74 | 51.41 | 57.77 | 49.83 | $53.20_{\pm0.28}$ |
| EMoE ($k'=12+2$) | 62.03 | 83.77 | 50.80 | 61.83 | 42.68 | 24.35 | 52.43 | 56.27 | 50.11 | $53.81_{\pm0.17}$ |
| *ERNIE-4.5-21B-A3B (trained with 6+2 activated experts)* | | | | | | | | | | |
| Top-$k$ ($k'=3+2$) | 88.81 | 94.18 | 75.44 | 80.86 | 66.46 | 26.18 | 59.10 | 64.17 | 71.58 | $69.64_{\pm0.15}$ |
| Top-$k$ ($k'=6+2$) | 89.15 | 95.24 | 79.83 | 81.56 | 71.34 | 26.43 | 61.10 | 66.93 | 72.86 | $71.60_{\pm0.23}$ |
| Top-$k$ ($k'=12+2$) | 88.81 | 94.89 | 80.52 | 80.08 | 68.29 | 26.81 | 60.24 | 67.09 | 71.92 | $70.96_{\pm0.41}$ |
| EMoE ($k'=3+2$) | 88.47 | 94.00 | 76.50 | 83.74 | 72.56 | 24.88 | 60.08 | 64.80 | 72.04 | $70.79_{\pm0.18}$ |
| EMoE ($k'=6+2$) | 88.47 | 94.89 | 78.92 | 84.54 | 74.39 | 25.07 | 61.76 | 67.40 | 73.87 | $72.15_{\pm0.30}$ |
| EMoE ($k'=12+2$) | 89.49 | 94.89 | 81.05 | 83.32 | 76.22 | 25.62 | 60.46 | 68.11 | 73.33 | $72.50_{\pm0.32}$ |

routing methods: AdaMoE (Zeng et al., 2024) and Top-$p$ (Huang et al., 2024), which dynamically adjust the number of activated experts across tokens while keeping the total number of activated experts fixed. In contrast, our method allows the total number of activated experts to be flexibly adjusted according to computational budgets. More comprehensive details about the baseline methods, including their implementation and configurations, are provided in Appendix A.1.

**Training and Evaluation Data.** Following Hui et al. (2024), we construct a diverse instruction-tuning dataset comprising 50K samples spanning three domains: coding, mathematics, and general abilities. Specifically, the dataset incorporates Magicoder (Wei et al., 2023) for coding, Meta-MathQA (Yu et al., 2024) for mathematics, and SlimORCA (Lian et al., 2023) for general abilities. For evaluation, we assess model performance across a comprehensive suite of nine downstream benchmark datasets, covering knowledge, reasoning, coding, and open-domain question answering. More details about the evaluation datasets can be found in Appendix A.1.

**Implementation Details.** In the LoRA-based MoE scenario, we activate 2 experts per layer during training to align with the sparse activation pattern typically used in large-scale models. In the FFN-based scenario, we follow the original pretraining configurations of the respective models: OLMoE activates 8 experts per layer, while DeepSeekV2-Lite and ERNIE-4.5-21B activates 6 fine-grained experts and 2 shared experts per layer. All MoE models are trained for 4 epochs. The learning rate is set to $2 \times 10^{-4}$ for LoRA-based settings and $2 \times 10^{-5}$ for FFN-based settings. All experiments are conducted three times, and we report the average results along with the standard deviation. More details about the hyperparameters can be found in Appendix A.1.

Table 2: Comparisons between EMoE and dynamic routing methods across different numbers of activated experts ($k'$) at inference. All methods are trained with a training budget equivalent to that of a standard Top-$k$ MoE with $k_{\text{train}} = 2$. For AdaMoE and Top-$p$, $k'$ refers to the average number of activated experts across all tokens.

| | $k'$ | ARC-c | ARC-e | GSM8K | HellaS. | HumanE. | NQ | Tri.QA | Wino | MMLU | AVG |
|---|---|---|---|---|---|---|---|---|---|---|---|
| Top-$k$ | 1.0 | 56.95 | 73.55 | 33.33 | 52.71 | 15.97 | 22.89 | 54.32 | 53.59 | 46.04 | $45.48_{\pm 0.91}$ |
| | 2.0 | 56.54 | 75.98 | 37.54 | 54.87 | 20.24 | 25.55 | 57.62 | 55.06 | 47.55 | $47.88_{\pm 0.57}$ |
| | 4.0 | 56.75 | 76.65 | 38.16 | 53.07 | 19.39 | 26.91 | 59.06 | 55.19 | 47.23 | $48.05_{\pm 0.36}$ |
| | 6.0 | 57.74 | 75.60 | 35.94 | 50.76 | 18.70 | 27.17 | 59.53 | 55.49 | 47.20 | $47.57_{\pm 0.52}$ |
| Top-$p$ | 1.0 | 56.95 | 74.25 | 35.56 | 49.19 | 15.85 | 24.60 | 55.58 | 56.43 | 46.41 | $46.09_{\pm 0.33}$ |
| | 2.2 | 55.59 | 77.60 | 36.62 | 50.14 | 17.07 | 25.79 | 57.67 | 55.88 | 47.00 | $47.04_{\pm 0.72}$ |
| | 4.1 | 59.66 | 79.01 | 36.92 | 49.54 | 20.12 | 27.06 | 59.41 | 54.30 | 46.88 | $48.10_{\pm 1.23}$ |
| | 6.0 | 56.95 | 78.84 | 36.92 | 46.48 | 20.12 | 27.12 | 60.14 | 53.51 | 47.25 | $47.48_{\pm 0.69}$ |
| AdaMoE | 1.3 | 55.93 | 67.20 | 34.12 | 49.83 | 18.29 | 14.74 | 35.95 | 52.17 | 43.83 | $41.34_{\pm 1.42}$ |
| | 2.2 | 60.68 | 75.13 | 37.30 | 55.28 | 20.73 | 22.58 | 52.03 | 53.99 | 46.89 | $47.18_{\pm 0.72}$ |
| | 4.2 | 58.31 | 77.25 | 37.68 | 56.30 | 20.12 | 27.81 | 58.63 | 54.70 | 46.06 | $48.54_{\pm 0.31}$ |
| | 6.1 | 56.95 | 77.07 | 37.60 | 56.11 | 20.73 | 28.95 | 59.25 | 54.30 | 45.76 | $48.52_{\pm 0.40}$ |
| EMoE | 1.0 | 55.25 | 71.78 | 31.24 | 51.52 | 17.68 | 25.46 | 54.79 | 56.75 | 46.64 | $45.68_{\pm 1.19}$ |
| | 2.0 | 56.95 | 78.66 | 37.98 | 53.57 | 18.90 | 26.62 | 58.20 | 55.41 | 47.65 | $48.22_{\pm 0.62}$ |
| | 4.0 | 58.31 | 79.72 | 38.21 | 55.95 | 18.29 | 27.78 | 59.24 | 55.64 | 47.89 | $49.00_{\pm 0.70}$ |
| | 6.0 | 60.34 | 79.37 | 38.21 | 56.14 | 20.73 | 27.87 | 59.77 | 55.33 | 47.73 | $49.50_{\pm 0.65}$ |

## 5.2 Main Results

**Comparisons to the Top-$k$ Method on Different Models.** Table 1 presents a comprehensive evaluation of our proposed EMoE framework against the standard Top-$k$ approach across three different model settings. These results align with our observations in Section 3.2. Across all models, we consistently observe performance degradation when the number of activated experts at inference exceeds the training budget. For example, using the standard Top-$k$ method achieves an average performance of 51.07 on DeepSeekV2-Lite, when the model is trained with $k_{\text{train}} = 6 + 2$. However, increasing the number of activated experts to $k = 12 + 2$ during inference results in a performance drop to 50.94. In contrast, models trained with the EMoE framework exhibit robust and monotonically increasing performance scalability. For every model architecture, increasing the number of activated experts at inference consistently leads to performance gains, confirming the effectiveness of the co-activation sampling. Notably, EMoE not only eliminates the performance drop observed in baselines but also leverages the proposed hierarchical loss to deliver further improvements under varying computational budgets, ultimately reaching new peaks in performance.

**Comparisons to Dynamic Routing Methods.** In Table 2, we further analyze EMoE and compare it with mainstream dynamic routing strategies. These methods are designed to optimize computational resource allocation under a fixed global computation budget by reallocating experts from simpler tokens to more complex ones. The results show that although these dynamic methods do provide some improvements over the static Top-$k$ baseline, for example, AdaMoE achieves a higher average performance of 48.54, and Top-$p$ performs best at low activation levels ($k' = 1$), but they ultimately still face the same issue of performance degradation. Their performance either plateaus or begins to degrade after reaching a peak, because these approaches are fundamentally not designed to go beyond a fixed computational limit. In contrast, EMoE demonstrates a distinctly superior scaling trend. Its performance increases monotonically as the number of activated experts grows, starting from a competitive baseline and eventually reaching the highest average score at $k' = 6$. This highlights a key distinction: prior methods focus on optimal reallocation under a fixed compute budget, whereas EMoE is uniquely designed to efficiently utilize variable and scalable computational resources.

## 5.3 Analysis

**Ablation Study.** Table 3 presents the individual contributions of the two key designs in EMoE: stochastic co-activation sampling and hierarchical router loss. The variant without stochastic co-activation sampling performs well when the number of experts is small, achieving the highest score of 45.83 at $k' = 1$. This strong performance can be attributed to the proposed router loss, which encourages a more hierarchical ranking among route experts and is particularly beneficial

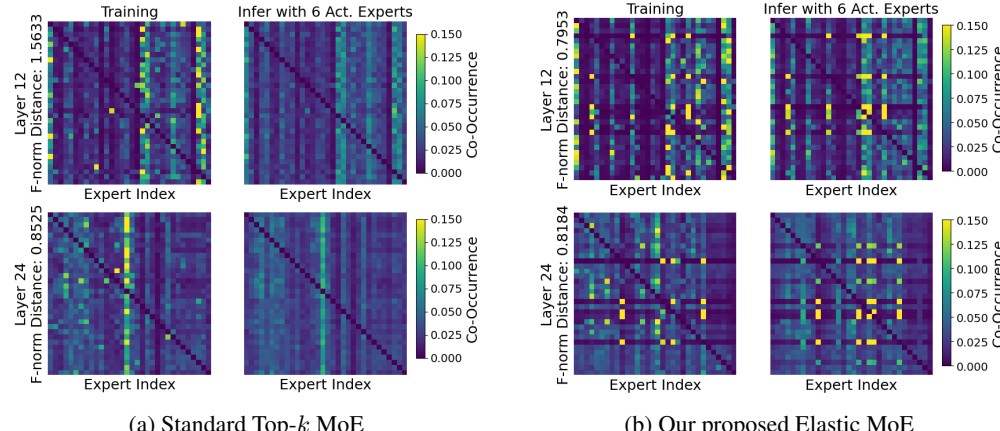

(a) Standard Top-$k$ MoE                   (b) Our proposed Elastic MoE

Figure 6: Visualization of expert co-occurrence matrices for (a) the standard Top-$k$ baseline and (b) our proposed EMoE. We compare the training pattern with inference using $k' = 6$, and report the F-norm distance between the corresponding matrices during training and inference.

under constrained computational budgets. However, as $k'$ increases to 6, the performance of this variant drops sharply. This result demonstrates that, without co-activation sampling, the model lacks effective collaboration between experts, ultimately leading to the collaboration collapse problem.

On the other hand, the variant without $\mathcal{L}_{\text{HR}}$ effectively alleviates the performance degradation at $k' = 6$, achieving a high score of 49.08. This confirms that co-activation sampling successfully facilitates collaboration among a wider set of experts. Nevertheless, this variant consistently underperforms the full EMoE model across all configurations and exhibits its lowest score at $k' = 1$. These results underscore the critical role of the proposed router loss in establishing a stable and hierarchical ranking of experts.

Table 3: Ablation study on EMoE's two key designs: stochastic co-activation sampling (co-act.) and the hierarchical router loss ($\mathcal{L}_{\text{HR}}$), compared with the full EMoE framework and the standard Top-$k$ baseline.

|  | $k' = 1$ | $k' = 2$ | $k' = 4$ | $k' = 6$ |
|---|---|---|---|---|
| Top-$k$ | 45.48 | 47.88 | 48.05 | 47.57 |
| EMoE | 45.68 | 48.22 | 49.00 | 49.50 |
| w/o co-act. | 45.83 | 48.03 | 48.68 | 48.15 |
| w/o $\mathcal{L}_{\text{HR}}$ | 45.19 | 47.79 | 48.81 | 49.08 |

Overall, these results demonstrate that both designs are essential, and only their combination allows the model to fully realize its potential for scalable performance at inference time.

**Effect of $k_{\text{ideal}}$.** We conduct analysis on the key hyperparameter $k_{\text{ideal}}$ in the co-activation sampling to verify the robustness of its configuration. The results in Figure 5 clearly demonstrate that the choice of $k_{\text{ideal}}$ is rather flexible and relaxed. Compared to the standard Top-$k$ baseline, even setting $k_{\text{ideal}}$ to just $2 \times k_{\text{train}} = 4$ yields significant performance gains, with both peak performance and low-budget performance surpassing the baseline. In particular, we observe that when $k_{\text{ideal}}$ is set between 2-4$\times$ the training expert count $k_{\text{train}}$, the model achieves optimal performance and scalability, reaching the highest average scores within this interval. On the other hand, when $k_{\text{ideal}}$ is set too high (e.g., exceeding $4 \times k_{\text{train}} = 8$), a trade-off emerges: while the model maintains strong performance under high inference budgets, its

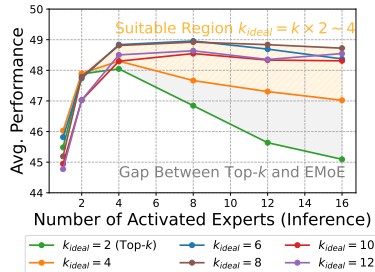

Figure 5: Analysis of the effect of the hyperparameter $k_{\text{ideal}}$. All experiments are conducted with $k_{\text{train}} = 2$.

performance with a low number of activated experts (such as $k' = 1$), as well as its overall peak performance, begin to degrade. Analysis on DeepSeekV2-Lite further confirms the validity of this relaxed range, as shown in Appendix A.2. Based on this analysis, we choose $k_{\text{ideal}} \in \{2, 3, 4\}$, using $4 \times k_{\text{train}}$ for the LoRA-based models, $3 \times k_{\text{train}}$ for DeepSeekV2-Lite and $2 \times k_{\text{train}}$ for OLMoE.

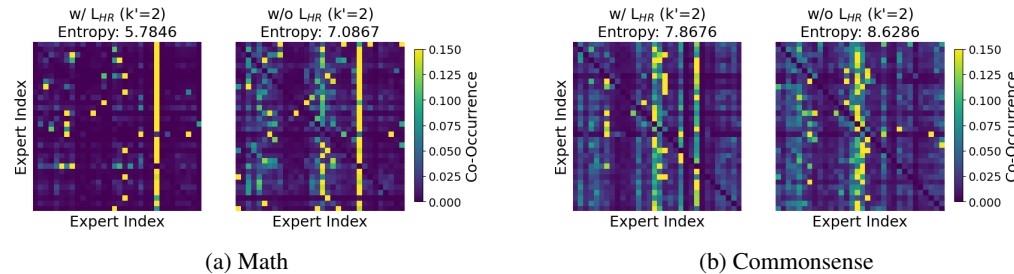

(a) Math            (b) Commonsense

Figure 7: Expert co-occurrence visualization under a low activation budget ($k' = 2$) across two domains. Across domains, models **with $\mathcal{L}_{\mathbf{HR}}$** exhibit sparse, concentrated hotspots, while models **without $\mathcal{L}_{\mathbf{HR}}$** show diffuse patterns with higher uncertainty.

**Effect of Stochastic Co-activation Sampling.** We visualize the expert co-occurrence matrix in Figure 6 and compare our method with the standard Top-$k$ method. When the EMoE trained model is extrapolated to $k' = 6$ during inference, co-activation sampling enables the co-occurrence matrix to maintain a high structural similarity to that observed during training. This stability is quantitatively supported by a sharply reduced Frobenius norm distance; for example, the F-norm distance at the 12th layer is only 0.79 for EMoE, in stark contrast to 1.56 for the Top-$k$ method. These results indicate that co-activation sampling effectively learns the expert combination patterns required under higher budgets, thereby ensuring scalability during inference.

**Effect of $\mathcal{L}_{\mathbf{HR}}$ on Expert Selection Stability.** To verify the effect of $\mathcal{L}_{\mathbf{HR}}$ on selecting more favorable expert combinations, we further visualize expert co-occurrence matrices under a low activation budget ($k' = 2$) on two domains: math and commonsense, using GSM8K (Cobbe et al., 2021) and HellaSwag (Zellers et al., 2019) respectively. As shown in Figure 7, models *without $\mathcal{L}_{\mathbf{HR}}$* exhibit diffuse, weakly structured co-activation patterns, indicating unstable selection when only a few experts can be used. In contrast, models *with $\mathcal{L}_{\mathbf{HR}}$* display sparse, concentrated hotspots, evidence of more decisive and consistent expert selection. This sharpening effect is also reflected quantitatively: entropy decreases from 7.09 to 5.78 (math) and from 8.63 to 7.87 (commonsense). These results show that $\mathcal{L}_{\mathbf{HR}}$ significantly stabilizes routing and is a key contributor to EMoE's elasticity.

**Scaling EMoE with Larger Training Budgets** To study whether EMoE continues to benefit from additional training compute, we further vary the training-time activation budget $k_{\mathrm{train}}$ and compare models trained with $k_{\mathrm{train}} \in \{2, 3, 4\}$. For each setting, we evaluate the resulting EMoE model under multiple inference budgets $k'$ and report the average performance across the same evaluation suite as in the main experiments. Figure 8 summarizes the results. We observe two trends. First, for each $k_{\mathrm{train}}$, EMoE maintains an elastic regime in which performance improves smoothly as $k'$ increases beyond $k_{\mathrm{train}}$. Second, increasing $k_{\mathrm{train}}$ consistently lifts the entire curve, especially at larger inference budgets. This shows that EMoE continues to benefit from additional training compute, while preserving its elasticity across $k'$.

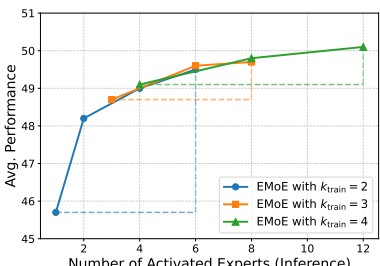

Figure 8: Average performance of EMoE for different training-time budgets $k_{\mathrm{train}} \in \{2, 3, 4\}$.

## 6   CONCLUSION

In this paper, we identify that existing MoE models suffer from performance degradation when scaling activated experts at inference due to insufficient expert collaboration. To address this issue, we propose Elastic MoE (EMoE), a training framework that enables flexible expert scaling without additional training overhead. EMoE incorporates stochastic co-activation sampling to foster expert collaboration and a hierarchical router loss to ensure stable expert selection. Extensive experiments show that EMoE exhibits robust, monotonically increasing performance as the inference budget grows, consistently outperforming the baseline at each budget level and surpassing its peak performance.

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

# A  APPENDIX

## A.1  IMPLEMENTATION DETAILS

**Details of Baselines.**  We evaluate our proposed EMoE framework against two primary categories of baselines: standard Top-$k$ routing and dynamic routing methods.

- **Standard Top-$k$ Routing:** This is the most prevalent approach in mainstream MoE models. Crucially, our EMoE framework is trained with the exact same number of activated experts to ensure an identical training overhead.
  - For the **FFN-based** MoE models, we adhere to the official configurations of OLMoE-0924 and DeepSeek-V2-Lite as used during their pre-training.
  - For the **LoRA-based** MoE, we adopt the sparse activation pattern commonly used in large-scale models (DeepSeek-AI et al., 2025), activating 2 out of 32 total experts (a 6.25% activation rate).
- **Dynamic Routing Methods:** We compare against two state-of-the-art dynamic routing techniques, Top-p and AdaMoE, which adjust expert activation per token.
  - **Top-$p$ Routing** (Huang et al., 2024) activates the smallest set of experts whose cumulative probability mass exceeds a threshold $p$. To maintain a comparable training budget, we set $p = 0.15$ during training. At inference, to match the average expert counts of other methods, we use $p$ values of $\{0.05, 0.16, 0.25, 0.34\}$.
  - **AdaMoE** (Zeng et al., 2024) introduces null experts that can be routed to, effectively allowing the model to skip computation for certain tokens. Following the original implementation, we set the number of null experts to be twice that of the standard experts. For a fair inference-time comparison, we vary its target active expert count $k'$ to $\{3, 6, 14, 22\}$ to align its average number of activated non-null experts with the computational budgets of Top-$k$ and EMoE.

**Details of Training Hyperparameters.**  In our main experiments, the learning rate is set to $2 \times 10^{-4}$ for all methods under the LoRA-based settings, and $2 \times 10^{-5}$ under the FFN-based settings. We use a batch size of 128 in all cases. All models are fine-tuned for 4 epochs on the dataset with a sequence length of 2048. For the hierarchical loss coefficient $\lambda$, we use a value of $5 \times 10^{-4}$ in LoRA-based scenarios and $1 \times 10^{-8}$ in FFN-based scenarios. Experiments are performed on 8 Nvidia H100 GPUs, each equipped with 80GB of memory. Every experiment is repeated three times, and we report the mean and standard deviation of the results.

**Details of Evaluation.**  We conduct a comprehensive evaluation utilizing the OpenCompass package (Contributors, 2023) to assess model performance across a diverse suite of downstream benchmarks. We report zero-shot accuracy on the commonsense and multitask reasoning tasks ARC-e, ARC-c (Clark et al., 2018), MMLU (Hendrycks et al., 2021), and WinoGrande (Sakaguchi et al., 2020). For reasoning capabilities, we measure 8-shot accuracy on the mathematical reasoning benchmark GSM8K (Cobbe et al., 2021) and 3-shot accuracy on HellaSwag (Zellers et al., 2019). Coding is evaluated via the pass@1 metric on HumanEval (Chen et al., 2021). To complete the assessment, our evaluation also includes two prominent open-domain question-answering benchmarks, Natural Questions (Kwiatkowski et al., 2019) and TriviaQA (Joshi et al., 2017).

## A.2  EXTENDED EXPERIMENTS

**Training Efficiency**  An essential consideration for the EMoE framework is its computational efficiency during the training process. To quantify its overhead, we compare it with the standard Top-$k$ baseline method. As shown in Table 4, EMoE achieves the same training overhead as the Top-$k$ baseline. This is because the core components of EMoE are introduced in the non-dense computation part of the computation graph. Specifically, both the sampling of experts from a larger candidate pool $\mathcal{S}_{k_{\text{ideal}}}(x)$ and the calculation of the hierarchical loss incur negligible computational cost compared to the dense matrix operations in Transformer models. Overall, EMoE successfully unlocks elastic scalability during inference with no extra training overhead, demonstrating its practical value as a lightweight and efficient training framework.

Table 4: Training overhead comparison between EMoE and the Top-$k$ baseline under the LoRA-based settings used in our main experiments. Both methods are trained with $k_{\text{train}} = 2$ on the same hardware.

| Method | $k_{\text{train}}$ | Training Time | Memory Usage Per GPU |
|--------|-----------|---------------|----------------------|
| EMoE | 2 | 10.92h | 43.4GB |
| Top-$k$ | 2 | 10.92h | 43.4GB |

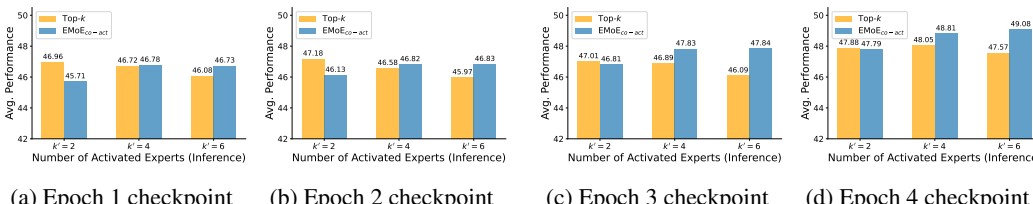

| (a) Epoch 1 checkpoint | (b) Epoch 2 checkpoint | (c) Epoch 3 checkpoint | (d) Epoch 4 checkpoint |
|---|---|---|---|

Figure 9: Performance evolution of EMoE$_{\text{co-act}}$ (i.e., only using co-activation sampling) versus the Top-$k$ baseline at different training checkpoints. The subplots (a) through (d) show performance snapshots at the end of epochs 1, 2, 3, and 4, respectively.

**Training Dynamics.** To gain deeper insight into the learning process of EMoE and investigate the impact of co-activation sampling on performance, we evaluate the model's performance at the end of each training epoch and compare it to the Top-$k$ baseline. Figure 9 illustrates this dynamic process. The experiments reveal a key trade-off. In the early stage of training (epoch 1), when only a small number of experts are activated during inference ($k' = 2$), EMoE with co-activation sampling only (i.e., EMoE$_{\text{co-act}}$) underperforms the Top-k baseline. We believe this is because the random sampling mechanism forces the model to explore a broader range of expert combinations, thus dispersing learning resources away from optimizing the most frequent Top-2 combinations. This leads to slightly slower convergence in this specific setting. However, even at this stage, our method already begins to outperform the Top-$k$ method in broader activation regimes ($k' = 4$ and $k' = 6$), indicating that the model has started to learn how to leverage more experts in collaboration.

As training progresses, this early trade-off is perfectly resolved. From epoch 2 onwards, EMoE$_{\text{co-act}}$ consistently matches or surpasses the baseline across all inference configurations. By the end of training (epoch 4), both models converge to their optimal performance, but with markedly different results. EMoE$_{\text{co-act}}$ achieves optimal performance under all inference budgets: not only does it match the Top-$k$ performance in the standard $k' = 2$ setting, but more importantly, it successfully extends this optimization to all activation regimes, exhibiting strong and monotonic performance scalability. In contrast, although the Top-$k$ baseline also converges to its optimal performance at $k' = 2$, its performance curve demonstrates that it fails to learn how to utilize additional experts.

**Effect of $k_{\text{ideal}}$ on DeepSeekV2-Lite.** We conduct an analysis of the hyperparameter $k_{\text{ideal}}$ on the DeepSeekV2-Lite model to further validate the robustness of its configuration. The results in Figure 10 clearly demonstrate that the choice of $k_{\text{ideal}}$ offers considerable flexibility and tolerance. Consistent with the conclusions drawn from Figure 5, setting $k_{\text{ideal}}$ to only twice the number of training experts ($k_{\text{train}} = 12 + 2$) leads to significant improvements in the performance of EMoE, exceeding the standard Top-$k$ in both peak and low-budget scenarios. Furthermore, when $k_{\text{ideal}}$ is set within 2 to 4 times the number of training experts, the model achieves optimal performance and scalability, reaching the highest average scores within this range.

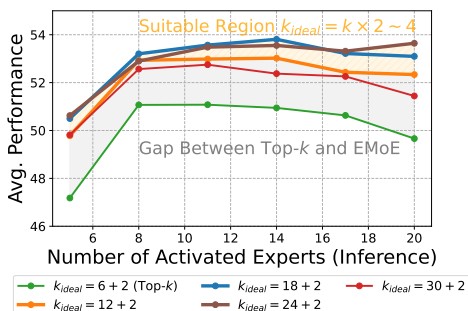

Figure 10: Analysis of the effect of the hyperparameter $k_{\text{ideal}}$. All experiments are conducted with $k_{\text{train}} = 6 + 2$.

Table 5: Comparison between EMoE and Top-$k$ trained with large $k_{\text{train}}$.

| Model | $k' = 1$ | $k' = 2$ | $k' = 4$ | $k' = 6$ | Mem / GPU |
|---|---|---|---|---|---|
| Top-$k$ ($k_{\text{train}} = 6$) | 42.87 | 46.26 | 48.30 | 49.14 | 76.0GB |
| EMoE (w/o $\mathcal{L}_{\text{HR}}$, $k_{\text{train}} = 2$) | 45.19 | 47.79 | 48.81 | 49.08 | 43.4GB |
| EMoE (with $\mathcal{L}_{\text{HR}}$, $k_{\text{train}} = 2$) | 45.68 | 48.22 | 49.00 | 49.50 | 43.4GB |

Table 6: Comparison between EMoE and standard Top-$k$ across inference-time budgets $k'$ on larger instruction-tuning datasets (100K and 200K samples). For each dataset size, both methods are trained under the same $k_{\text{train}} = 2$.

| | ARC-c | ARC-e | GSM8K | HellaS. | HumanE. | NQ | Tri.QA | Wino. | MMLU | AVG |
|---|---|---|---|---|---|---|---|---|---|---|
| *100K instruction data* | | | | | | | | | | |
| Top-$k$ ($k' = 1$) | 53.90 | 71.25 | 34.19 | 58.47 | 20.12 | 21.99 | 52.70 | 55.41 | 46.05 | 46.01 |
| Top-$k$ ($k' = 2$) | 54.24 | 75.66 | 42.00 | 56.67 | 25.61 | 24.82 | 57.49 | 54.38 | 46.66 | 48.61 |
| Top-$k$ ($k' = 4$) | 55.93 | 77.78 | 43.75 | 55.53 | 26.83 | 27.37 | 59.14 | 55.49 | 45.38 | 49.69 |
| Top-$k$ ($k' = 6$) | 56.27 | 77.25 | 40.33 | 53.51 | 21.34 | 28.42 | 59.88 | 54.62 | 46.45 | 48.67 |
| EMoE ($k' = 1$) | 57.29 | 76.54 | 37.45 | 57.01 | 14.02 | 24.07 | 54.76 | 53.51 | 46.57 | **46.80** |
| EMoE ($k' = 2$) | 60.00 | 79.72 | 40.64 | 59.47 | 17.07 | 26.40 | 58.45 | 55.33 | 47.86 | **49.44** |
| EMoE ($k' = 4$) | 62.71 | 80.95 | 41.09 | 61.04 | 20.12 | 27.09 | 59.57 | 55.56 | 48.86 | **50.78** |
| EMoE ($k' = 6$) | 63.05 | 80.95 | 42.30 | 61.13 | 21.34 | 27.42 | 60.09 | 56.20 | 48.87 | **51.26** |
| *200K instruction data* | | | | | | | | | | |
| Top-$k$ ($k' = 1$) | 58.31 | 76.54 | 40.03 | 61.26 | 27.44 | 23.13 | 53.92 | 52.41 | 48.35 | 49.04 |
| Top-$k$ ($k' = 2$) | 60.34 | 79.37 | 44.66 | 63.92 | 29.88 | 25.60 | 58.09 | 53.20 | 50.09 | 51.68 |
| Top-$k$ ($k' = 4$) | 61.02 | 81.13 | 44.96 | 63.46 | 28.66 | 27.40 | 59.91 | 54.14 | 50.66 | 52.37 |
| Top-$k$ ($k' = 6$) | 59.32 | 81.31 | 43.52 | 62.11 | 25.61 | 27.34 | 60.29 | 53.67 | 50.27 | 51.49 |
| EMoE ($k' = 1$) | 62.71 | 75.13 | 42.30 | 60.98 | 21.95 | 25.04 | 56.08 | 55.33 | 47.51 | **49.67** |
| EMoE ($k' = 2$) | 65.08 | 78.84 | 44.28 | 62.31 | 25.61 | 25.57 | 58.67 | 58.56 | 49.19 | **52.01** |
| EMoE ($k' = 4$) | 64.75 | 81.31 | 46.47 | 63.29 | 22.56 | 27.70 | 60.20 | 58.17 | 50.21 | **52.74** |
| EMoE ($k' = 6$) | 66.44 | 78.13 | 47.38 | 63.60 | 25.61 | 27.98 | 60.36 | 57.14 | 51.99 | **53.18** |

**Comparison to Training with Larger $k_{\text{train}}$.** We compare EMoE trained with $k_{\text{train}} = 2$ against a standard Top-$k$ MoE trained with $k_{\text{train}} = 6$. The large-$k_{\text{train}}$ model achieves only a marginal advantage than EMoE without $\mathcal{L}_{\text{HR}}$ at its native inference budget ($k' = 6$), but exhibits two clear drawbacks: (1) Lack of flexibility. Performance degrades sharply when reducing the inference budget (e.g., $k' = 2 < k_{\text{train}}$), showing that standard MoE training strongly couples the router to the training-time budget. (2) Excessive training cost. beyond the standard $k$ configuration (e.g., from 2 to 6) requires roughly $3\times$ FLOPs and more activation memory in MoE layers, making such large-$k$ training impractical at scale.

In contrast, **EMoE maintains strong performance under both lower inference budgets ($k' < k_{\text{train}}$) and higher inference budgets ($k' > k_{\text{train}}$),** while retaining the training cost of standard Top-$k$ models. Thus, EMoE provides inference-time elasticity that large-$k_{\text{train}}$ training is unable to offer and eliminates the need to deploy multiple MoE variants for different computational budgets.

**Scalability Analysis on Larger Instruction-Tuning Datasets.** We extend our experiments to larger 100K and 200K instruction datasets. The results are reported in Table 6. Across both data scales, EMoE consistently preserves its elastic window and maintains strong extrapolation capability beyond the training-time activation budget. Importantly, enlarging the dataset does not diminish the benefits of EMoE: the method continues to outperform standard Top-$k$ training at both lower ($k' < k_{\text{train}}$) and higher ($k' > k_{\text{train}}$) inference budgets. These results show that EMoE's effectiveness is independent of dataset scale and, as confirmed by our main experiments, can already be achieved efficiently with a lightweight 50K instruction set.

**Analysis of Expert Diversity.** To examine how the two components in our method influence expert specialization, we measure the mutual information (MI) between expert selection and task domains (math: GSM8K (Cobbe et al., 2021), commonsense: HellaSwag (Zellers et al., 2019), code: HumanEval (Chen et al., 2021)). Let $P(e)$ denote the marginal expert usage and $P(e \mid d)$ the

domain-conditional usage. We compute:

$$\text{MI}(E; D) = \sum_{e,d} P(e, d) \log \frac{P(e, d)}{P(e)P(d)}, \tag{9}$$

where $P(e, d) = P(e \mid d)P(d)$. As shown in table 7, co-activation sampling is the primary factor that enhances specialization by exposing experts to diverse collaborative configurations during training. A higher degree of specialization indicates that experts consistently assume distinct functional roles across domains, which is an important symptom of successful collaborative organization rather than redundant or interchangeable behaviors. The hierarchical router loss further improves this structure by producing more decisive expert rankings, achieving the highest MI. These findings show that $\mathcal{L}_{\text{HR}}$ works synergistically with co-activation sampling and plays a central role in EMoE's elasticity.

Table 7: Mutual information between expert usage and task domain.

| Model Setup | MI | $\Delta$ |
|---|---|---|
| Baseline (Standard Top-$k$) | 0.0473 | – |
| + Co-activation Sampling (w/o $\mathcal{L}_{\text{HR}}$) | 0.0603 | +27.5% |
| + Full EMoE (with $\mathcal{L}_{\text{HR}}$) | **0.0630** | +4.5% |

## A.3 ALGORITHM OF EMOE

Here, we present the complete algorithm of the proposed EMoE training framework in Algorithm 1.

---

**Algorithm 1** Elastic Mixture-of-Experts (EMoE) Training Framework

---

1: **Require:** Input $x$, Router $G$, Experts $\{E_i(\cdot; \theta_i)\}_{i=1}^{N}$
2: **Hyperparameters:** $k_{\text{ideal}}, \lambda$

3: **Step 1: Get router logits**
4: $h(x) \leftarrow G(x)$                  $\triangleright$ Raw logits for all experts, $h(x) \in \mathbb{R}^N$

5: **Step 2: Stochastic co-activation sampling**
6: **Step 2a: Determine candidate pool size**
7: $\tilde{k}_{\text{ideal}} \sim \text{UniformInt}(k_{\text{train}}, k_{\text{ideal}})$
8: $\mathcal{S}_{\tilde{k}_{\text{ideal}}}(x) \leftarrow \text{TopKIndices}(h(x), \tilde{k}_{\text{ideal}})$      $\triangleright$ Select candidate experts based on top logits
9: **Step 2b: Sample experts for forward pass**
10: $\mathcal{S}_{\text{co-act}}(x) \sim \text{UniformSample}(\mathcal{S}_{\tilde{k}_{\text{ideal}}}(x), k_{\text{train}})$      $\triangleright$ Final subset used for training

11: **Step 3: Compute MoE output**
12: $y_{\text{co-act}}(x) \leftarrow \sum_{i \in \mathcal{S}_{\text{co-act}}(x)} \frac{\exp(h_i(x))}{\sum_{j \in \mathcal{S}_{\text{co-act}}(x)} \exp(h_j(x))} \cdot E_i(x; \theta_i)$

13: **Step 4: Compute total loss**
14: $\mathcal{L}_{\text{ce}} \leftarrow \text{CrossEntropyLoss}(y_{\text{co-act}}(x), \text{target})$
15: $\mathcal{L}_{\text{b}} \leftarrow \text{LoadBalancingLoss}(h(x))$
16: $\mathcal{L}_{\text{HR}} \leftarrow -\sum_{i=1}^{N} h_i(x) \log \frac{h_i(x)}{1/N}$      $\triangleright$ Hierarchical router loss (encourage decisive ranking)
17: $\mathcal{L}_{\text{total}} \leftarrow \mathcal{L}_{\text{ce}} + \mathcal{L}_{\text{b}} + \lambda \cdot \mathcal{L}_{\text{HR}}$
18: **return** $\mathcal{L}_{\text{total}}$

---

## A.4 BROADER IMPACT AND LIMITATIONS

**Broader Impact.** Our work introduces the Elastic Mixture-of-Experts (EMoE) framework, which aims to address a fundamental technical challenge: enabling scalable computation for large models during inference without increasing training costs. While this technical breakthrough is rooted in model architecture, its potential applications may have far-reaching implications for both the field

of artificial intelligence and society. The most immediate contribution of EMoE is lowering the barrier to high-performance model inference. By allowing users to dynamically adjust computational expenditure based on available hardware resources, EMoE makes state-of-the-art models more accessible to a broader range of researchers, small and medium-sized enterprises, and independent developers, thereby promoting the democratization of AI technology. Furthermore, the flexible computation enabled by EMoE allows systems to switch to low-power modes (activating fewer experts) during off-peak periods or for simpler tasks. This adaptability can significantly reduce overall energy consumption in large-scale deployments, contributing to the development of a greener and more sustainable AI ecosystem.

**Limitations.** Although our study provides strong evidence for the effectiveness of the Elastic Mixture-of-Experts (EMoE) framework across various model sizes and architectures, we acknowledge the following limitations in our current work: First, while EMoE significantly expands the scalability of MoE models during inference, this elastic range is not without boundaries. Our analysis of the hyperparameter $k_{\text{ideal}}$ in Figure 6 clearly reveals this inherent trade-off. We observe that when $k_{\text{ideal}}$ is set within the range of 2 to 4 times the number of training experts $k_{\text{train}}$, the model achieves optimal performance and scalability. However, setting $k_{\text{ideal}}$ excessively large may lead to diminishing returns. Specifically, in the extreme case where $k_{\text{ideal}}$ is set to the total number of experts $N$ while $k_{\text{train}}$ is 2, this is equivalent to randomly selecting two experts from all experts for training, which renders the router ineffective. This means that such a boundary naturally exists and cannot be extended indefinitely. We will explore approaches to further extend this effective range in future work.

Second, our validation on ultra-large-scale models is limited. Due to computational constraints, our experiments primarily focus on models ranging from 7B to 16B parameters. While EMoE demonstrates consistent and robust scalability across these sizes, suggesting promising generalization to even larger models, we have not yet directly evaluated it on models with over 100B parameters. Extending EMoE to such ultra-large models represents an important and promising direction for future research, and our current results provide encouraging preliminary evidence for this potential.

A.5 USE OF LLMS

We use large language models solely for the purpose of improving the grammar and language clarity of our manuscript. No LLMs are used for generating ideas, designing experiments or writing substantive content. All scientific contributions, results, and conclusions are entirely the work of the authors.

