# OpenReview forum: "Elastic MoE: Unlocking the Inference-Time Scalability of Mixture-of-Experts"
_ICLR.cc/2026/Conference — Submitted to ICLR 2026_

### Official Review · Reviewer_rgdX · 2025-10-31

**Soundness:** 3
**Presentation:** 3
**Contribution:** 3
**Rating:** 4
**Confidence:** 4

**Summary:**

The paper postulates that elastic allocation of compute in MoE layers (e.g., activating more experts per token than in the training setup) can provide additional flexibility for inference. However, as the authors show, the naive approach, where we just increase the top-k in the router, does not result in model quality improvement. The authors hypothesize that the reason is lack of collaboration between experts - the model is not able to leverage new combinations of experts not seen during training. The proposed solution is to "simulate" training with larger top-k by sampling smaller groups of experts from larger, "ideal" groups, that will be seen during inference. The paper shows that when this method is in use, increasing router top-k to ks larger than seen during training is significantly more effective.

**Strengths:**

1. The paper is clearly written and easy to follow.
2. The motivation is clear. The paper tackles a natural problem, provides convincing reasoning on its nature, and proposes a solution.
3. The methodology is consistent and the results are positive.
4. Evaluation is based on three open-source models and two training scenarios (LoRA and MoE FFN).

**Weaknesses:**

1. Examination of the proposed approach is limited to training on the small instruction-tuning dataset.
2. The instruction fine-tuning is extremely lightweight (only 50K samples). Let's say we have an MoE model trained with topk=2. If we want to infer it with topk=4, we could just directly fine-tune on top-k=4, instead of using co-activation sampling, where we sample 2 experts from groups of size 4. The argument of the reduced training cost with stochastic co-sampling is not relevant since the entire fine-tuning phase has negligible cost compared to the model pretraining. This weakness is connected to weakness 1. mentioned above.
3. The introduction of hierarchical router loss feels a bit orthogonal to the main body of the paper, especially given results in the Section 5.3, showing that the main ingredient contributing to the benefit of EMoE is co-activation sampling.

**Questions:**

1. Did authors consider evaluating the proposed technique in the pretraining or continued pretraining setup?
2. In Table 1, there is a comparison of models trained with k experts when using k'>k during inference. Could the authors add to the comparison the model fine-tuned on the target k'? So for example, in the second group of Table 1, I would love to see a comparison between EMoE(k'=16) and OLMoE-1B-7B-0924 directly fine-tuned on topk=16. This model directly fine-tuned with topk=16 would technically have a larger cost, but as mentioned before, any cost of fine-tuning on the small dataset can be considered negligible, so ultimately the best approach is the one giving the best performance after tuning on this dataset.
3. The authors may also comment or provide additional experiments on the points mentioned in the Weaknesses section.

---

> ### Author Response · Authors · 2025-11-18
> **Response to reviewer rgdX (1/3)**
>
> We sincerely thank Reviewer rgdX for the thorough review and positive evaluation of our manuscript. We greatly appreciate your recognition of the motivation, methodological framework, and experimental design of our work. The questions you raised are highly relevant and have helped us to clarify the positioning, advantages, and application scenarios of EMoE. During the revision process, we have conducted additional experiments and provided further clarifications to address these issues. Our detailed responses are as follows:
>
> ## W1: Evaluation limited to small instruction-tuning dataset
>
> In the revised version, we have supplemented our results with experiments conducted on **100K and 200K instruction datasets (see Table 6, Page 18 in the revised paper)** to examine whether the effectiveness of EMoE diminishes as the data scale increases. The experiments show that regardless of the extent to which the dataset size is enlarged, EMoE consistently maintains its elastic window (2–3×k) and extrapolation capability. This further demonstrates that the effectiveness of our method is not a coincidental outcome of small datasets, but rather a generalizable property.
>
> ## W2/Q2: Why not directly fine-tune with a larger top-k?
>
> ### (A) Large-k fine-tuning collapses under low-budget inference
>
> We fully agree that comparing EMoE to models directly trained with higher k values is essential for assessing the value of our approach. Here, we compare our EMoE model (trained with k_train=2) against a standard Top-k model trained directly with a larger number of experts (k_train=6) on the LoRAMoE setting.
>
> | Model (LoRAMoE) | k'=1 | k'=2 | k'=4 | k'=6 | Memory Cost (Per GPU) |
> | --- | --- | --- | --- | --- | --- |
> | Top-k (k_train=6) | 42.87 | 46.26 | 48.3 | 49.14 | 76.0GB |
> | EMoE (w/o $\mathcal{L}\_\text{HR}$, k_train=2) | 45.19 | 47.79 | 48.81 | 49.08 | 43.4GB |
> | EMoE (w/ $\mathcal{L}\_\text{HR}$, k_train=2) | 45.68 | 48.22 | 49.00 | 49.5 | 43.4GB |
>
> Similarly, on the OLMoE setting, we compare the performance of the EMoE model (trained with k_train=8, **inherited from the pre-training configuration**) to a larger Top-k model  (k_train=16).
> | Model (OLMoE) | k'=4 | k'=8 | k'=16 |
> | --- | --- | --- | --- |
> | Top-k (k_train=16) | 34.25 | 40.26 | 41.19 |
> | EMoE (w/ $\mathcal{L}\_\text{HR}$, k_train=8) | 37.36 | 42.05 | 42.58 |
>
> A large-k model performs reasonably only at its native training budget but **fails sharply at smaller k′**, which are essential in real deployments, **as they represent the conventional sparse activation mode in practical MoE systems**. By contrast, EMoE, trained with the conventional sparse configuration, **can be extrapolated at inference to settings that are 2–3 times larger than the training configuration**, while still maintaining effectiveness under low-budget settings.
>
> ### (B) Scaling Up k_train in EMoE for Further Gains
>
> In Figure 8 on page 10 of our revised manuscript, we additionally evaluate EMoE with higher training budgets (e.g., k_train=3,4). These experiments show EMoE continues to improve in the large-k′ regime as k_train increases.
>
> This demonstrates that:
>
> 1. EMoE’s performance is **not limited** by training with a single k_train.
> 2. EMoE **benefits from more compute**, while standard Top-k **loses flexibility** as k_train grows.
> 3. EMoE thus provides a strictly more favorable trade-off between training cost and inference flexibility.
>
> ### (C) Deployment requires one model that supports multiple inference budgets
>
> In real systems, a MoE model is often deployed across hundreds or thousands of GPUs. Retraining and redeploying a new model for each k′ setting is impractical, and deploying multiple large-parameter top-k models simultaneously is infeasible. Standard Top-k models inherently bind training and inference to the same k, causing degradation whenever k′ differs.
>
> EMoE is designed precisely for this scenario: a single model can reliably cover a wide range of budget points (both lower and higher than the training k), **without the need for retraining or deploying multiple versions.** Thus, the problem EMoE solves is not just “how to fine-tune more cheaply,” but rather: **how to ensure that a single MoE model can support multi-budget inference immediately post-deployment that large top-k training simply cannot provide.**

---

> ### Author Response · Authors · 2025-11-18
> **Response to reviewer rgdX (2/3)**
>
> ## W3 Motivation of $\mathcal{L}\_\text{HR}$
>
> We greatly appreciate your suggestion to more closely link the motivation for $\mathcal{L}\_\text{HR}$ with the central theme of “expert collaboration” in our paper. We have reinforced this connection in the revised manuscript. **The goal of EMoE is to deliver reliable performance across varying inference-time budgets while preserving the standard training cost of conventional k configurations**, which requires solving **two interrelated challenges**:
>
> 1. Teaching experts how to collaborate effectively in diverse group sizes and combinations, a capability fostered by our *Stochastic Co-activation Sampling*. This training endows experts with the necessary “collaborative skills” for functioning in variable inference budgets.
> 2. Teaching the router how to select the most favorable expert combinations under different computational budgets, precisely where the *Hierarchical Router Loss ($\mathcal{L}\_\text{HR}$)* plays a pivotal role.
>
> While stochastic co-activation equips experts with collaborative skills, the router needs to identify the most effective combinations, especially under small activation budgets. Without $\mathcal{L}\_\text{HR}$, this selection mechanism can be unstable. As shown in our ablation (Table 3 on page 9), the variant without $\mathcal{L}\_\text{HR}$ performs noticeably worse at low activation budgets (k′ = 1 and k′ = 2), revealing that **router instability becomes the bottleneck even when expert collaboration is available**. $\mathcal{L}\_\text{HR}$ is designed to directly address this issue.
>
> As shown in Figure 7 on page 10 of our revised paper, we compare the expert co-occurrence matrices during inference (**k’=2, denote low activation budgets**) for models with and without $\mathcal{L}\_\text{HR}$, across two specialized domains: mathematical reasoning (GSM8K) and commonsense reasoning (HellaSwag). This figure clearly demonstrates the impact of $\mathcal{L}\_\text{HR}$ on the router’s selection strategy:
>
> - The model without $\mathcal{L}\_\text{HR}$ (right panel) displays a relatively diffuse co-occurrence pattern. This indicates that, in the absence of the loss, the router can activate collaborative experts but lacks decisiveness and consistency in picking favorable pairs for the given task, resulting in uncertainty in its selection strategy.
> - In contrast, the model with $\mathcal{L}\_\text{HR}$ (left panel) exhibits a sparse, highly concentrated co-occurrence pattern dominated by a few pronounced “hotspots.” This evidences that it successfully guides the router to consistently select favorable expert combinations for specific tasks.
>
> This optimization of the selection strategy is also reflected in information entropy: the entropy of the co-occurrence matrix without $\mathcal{L}\_\text{HR}$ is 7.1, while with $\mathcal{L}\_\text{HR}$ it is reduced to 5.8 for the math domain and 8.6 vs 7.9 for the commonsense domain, indicating a **nearly 10%-20% decrease in uncertainty**. In summary, $\mathcal{L}\_\text{HR}$ is not an orthogonal add-on, but a **crucial component that stabilizes the router’s combinatorial selection under varying budgets**, particularly at low activation budgets, thereby enabling EMoE to achieve robust and truly elastic behavior.

---

> ### Author Response · Authors · 2025-11-18
> **Response to reviewer rgdX (3/3)**
>
> ## Q1: Have you considered validating EMoE during pre-training or incremental pre-training?
>
> We greatly appreciate the reviewer’s suggestion to validate EMoE during the pre-training or continual pre-training phase, and we agree that this could further demonstrate the method’s potential value. However, **EMoE** **is intentionally designed to avoid intrusive modifications to the pre-training process**, enabling the method to enhance inference elasticity in downstream training through a “plug-and-play, low-cost” approach.
>
> Because EMoE does not rely on structural changes during pre-training, it can be directly applied in SFT or continued-tuning stages. This property has important practical implications. In many real-world systems, MoE models are already fully pre-trained, and restarting pre-training to support a different inference budget is prohibitively expensive. **In such cases, the ability to obtain stable and controllable cross-budget behavior through a lightweight SFT procedure makes heavy pretraining generally unnecessary in practice, especially for large-scale MoE models.**
>
> In addition, we have validated the effectiveness of EMoE under two completely different training paradigms (i.e., LoRA-MoE and FFN-MoE), further demonstrating that the method does not depend on specific pre-training details, but can be generally applied to existing MoE models.
>
> Therefore, from the perspectives of practicality and integrability, the SFT stage itself is a highly representative intervention point, fully showcasing EMoE’s plug-and-play advantages.
>
> Thank you once again for your valuable feedback. We are committed to refining our work based on your insights.

---

> ### Author Response · Authors · 2025-11-24
>
> Dear Reviewer rgdX,
>
> Thank you again for your thoughtful and constructive reviews. We hope our responses have helped clarify the concerns raised, and we truly appreciate the opportunity to improve our work.
>
> If there are any remaining questions or suggestions, we would be grateful to hear them. Your feedback is incredibly valuable to us.
>
> ICLR 2026 Submission16399 Author

---

### Official Review · Reviewer_kgGr · 2025-10-31

**Soundness:** 3
**Presentation:** 3
**Contribution:** 2
**Rating:** 4
**Confidence:** 4

**Summary:**

The paper addresses the topic of improving the inference-time scalability of Mixture-of-Experts (MoE) models. The authors observe that standard MoE architectures, when trained with a fixed number of active experts (k), tend to perform poorly if a larger number (k′>k) is used at inference. They identify the cause as insufficient co-training of expert combinations and propose Elastic Mixture-of-Experts (EMoE) to mitigate this issue. EMoE combines (1) stochastic co-activation sampling, which encourages expert collaboration across diverse subsets without additional training cost, and (2) a hierarchical router loss that produces more stable expert rankings. The experiments show that EMoE models trained with small k exhibit improved performance when evaluated with higher k′, outperforming standard Top-k and dynamic routing baselines.

**Strengths:**

* The topic is very timely and important, as efficient scaling and inference-time flexibility in MoE models are of central interest to current LLM research.
* The paper is clearly written and presents the motivation and results in a structured way.
* The empirical evaluation is thorough and includes ablations isolating the effects of the proposed components.
* The approach introduces almost no additional training cost and could be easily adopted in existing MoE frameworks.

**Weaknesses:**

The main conceptual limitation is the lack of comparison with a model trained with a larger number of experts. Figure 2 in the paper itself shows that when a model is trained with higher k_train, it already generalizes better to larger k′ at inference. This raises an important baseline question: if we simply train the model for larger k, would that not achieve the same or better performance than EMoE, possibly without additional algorithmic complexity?

Since EMoE aims to emulate large-k behavior while training with small k, the natural control experiment would be: train standard MoE with k_train = 6 or 8, evaluate it at comparable inference budgets, and compare both training cost and resulting performance. Without this, it is difficult to assess whether EMoE’s advantage stems from genuine elasticity or from the baseline’s artificially small k. In practice, the training-cost versus performance trade-off determines real-world usefulness, and that dimension is underexplored.

**Questions:**

1. How does EMoE compare to a model trained directly with a higher k_train, both in performance and in compute cost?
2. Would the claimed benefits persist if training compute were scaled accordingly (i.e., fair total-FLOP comparison)?

---

> ### Author Response · Authors · 2025-11-18
> **Response to Reviewer kgGr (1/2)**
>
> We sincerely appreciate Reviewer kgGr for the thorough review and valuable comments on our work. We are very grateful for your recognition of the significance of our research topic, the clear structure of our manuscript, and the practical usability of our proposed method.  Below, we address each of your concerns in detail.
>
> ## W1/W2/Q1: Comparison to Top-k Models Trained with Larger k
>
> ### (A) Results of Top-k Models Trained with Larger k
>
> This is an excellent suggestion and a crucial baseline for our study. Here, we compare our EMoE model (trained with k_train=2) against a standard Top-k model trained directly with a larger number of experts (k_train=6).
>
> | Model | k'=1 | k'=2 | k'=4 | k'=6 | Memory Cost (Per GPU) |
> | --- | --- | --- | --- | --- | --- |
> | Top-k (k_train=6) | 42.87 | 46.26 | 48.3 | 49.14 | 76.0GB |
> | EMoE (w/o $\mathcal{L}\_\text{HR}$, k_train=2) | 45.19 | 47.79 | 48.81 | 49.08 | 43.4GB |
> | EMoE (w/ $\mathcal{L}\_\text{HR}$, k_train=2) | 45.68 | 48.22 | 49.00 | 49.50 | 43.4GB |
>
> The experimental results clearly demonstrate the trade-offs. At its native inference budget (k′ = 6), the k_train = 6 Top-k model is only slightly better than EMoE without $\mathcal{L}\_\text{HR}$, and performs on par with the full EMoE model. However, it suffers from two important limitations:
>
> 1. **Lack of flexibility:** When **reverting to the conventional sparse MoE configuration (e.g., k′ = 2)** for lower-cost inference, its **performance drops sharply**, making it unsuitable for dynamically changing deployment environments.
> 2. **Higher training cost:** Increasing k_train from 2 to 6 leads to **approximately 3× FLOPs and more activation memory** in the MoE layers during training, which grows increasingly significant as model size scales.
>
> ### (B) Our Intent Behind Small k_train Experiments
>
> Our experiments with **small** **k_train were not chosen deliberately; they reflect the standard expert-activation configuration used during pre-training (e.g. A6+2 for DeepSeek-V2-Lite and A8 for OLMoE).** The purpose of these experiments is to demonstrate that EMoE can be reliably extrapolated to larger k′ values while maintaining this conventional training setup.
>
> Thus, the intention is **not** to claim that increasing k_train is unnecessary, but rather to show: EMoE provides the advantages of large-k training while using far less compute, and without sacrificing performance at smaller inference budgets. This extrapolation ability is precisely what standard Top-k training fails to provide, as its behavior is tightly bound to the training-time expert count.
>
> ### (C) Scaling Up k_train in EMoE for Further Gains
>
> In Figure 8 on page 10 of our revised manuscript, we additionally evaluate EMoE with higher training budgets (e.g., k_train=3,4). These experiments show EMoE continues to improve in the large-k′ regime as k_train increases.
>
> This demonstrates that:
>
> 1. EMoE’s performance is **not limited** by training with a small k_train.
> 2. EMoE **benefits from more compute**, while standard Top-k **loses flexibility** as k_train grows.
> 3. EMoE thus provides a strictly more favorable trade-off between training cost and inference flexibility.
> ### (D) Deployment requires one model that supports multiple inference budgets
>
> In real systems, a MoE model is often deployed across hundreds or thousands of GPUs. Retraining and redeploying a new model for each k′ setting is impractical. Standard Top-k models inherently bind training and inference to the same k, causing degradation whenever k′ differs.
>
> EMoE is designed precisely for this scenario: a single model can reliably cover a wide range of budget points (both lower and higher than the training k), **without the need for retraining or deploying multiple versions.** Thus, the problem EMoE solves is not just “how to fine-tune more cheaply,” but rather: **how to ensure that a single MoE model can support multi-budget inference conveniently.**

---

> ### Author Response · Authors · 2025-11-18
> **Response to Reviewer kgGr (2/2)**
>
> ## Q2. Would the benefits persist if the training compute is scaled?
>
> Yes. We examined this question from **two complementary scaling angles**:
>
> ### (A) Scaling up the *training-time activation budget* (larger k_train)
>
> As shown in the new experiments included in Figure 8 of our revised manuscript, when we increase EMoE’s own training-time activation budget (e.g., from k_train = 2 to 3 and 4), EMoE continues to improve, especially in the large-k′ inference regime.
>
> This confirms that EMoE is not merely a compute-efficient alternative for small budgets; rather, it is a **scalable mechanism** that keeps benefiting from additional training compute.
>
> ### (B) Scaling up the *model size* (larger parameter count)
>
> In the main experiments, EMoE is validated across models from 7B-scale to 16B-scale. In the revision, we further include a substantially larger **21B-scale MoE model (ERNIE-4.5-21B-A3B) in Table 1 on page 7**. Despite the significant increase in capacity and the model’s native training with 6+2 active experts, we observe the same pattern: standard Top-k behaves well only near its training-time activation setting, while EMoE remains stable or improves when k′ is decreased or increased. The elasticity behavior seen in smaller models reproduces almost identically at 21B scale.
>
> Taken together, these results confirm that EMoE’s advantages persist under both **training-compute scaling** and **model-size scaling**. EMoE is not only efficient at small budgets but also remains effective and increasingly advantageous as compute and model capacity grow.
>
> Thank you once again for your valuable feedback. We are committed to refining our work based on your insights.
>
> If there are any remaining concerns or questions that we have not yet addressed, please do not hesitate to let us know. We are more than willing to provide any additional materials or clarifications that may be necessary.

---

> > ### Comment · Reviewer_kgGr · 2025-11-19
> >
> > Thank you for the explanation, I appreciate the additional experiments and context. Based on your arguments I am raising my score.

---

> ### Author Response · Authors · 2025-11-20
> **Thank you for Reviewer's reply**
>
> Dear Reviewer kgGr,
>
> We are very glad to hear that our additional experiments and clarifications have addressed your concerns. Thank you for raising your score to 6. Your feedback and suggestions were highly valuable to improving the quality and clarity of our work.
>
> ICLR 2026 Submission16399 Author

---

### Official Review · Reviewer_UsPv · 2025-11-04

**Soundness:** 2
**Presentation:** 3
**Contribution:** 2
**Rating:** 4
**Confidence:** 3

**Summary:**

The paper proposes Elastic Mixture-of-Experts (EMoE), a training strategy that enables post-hoc inference scalability in sparse MoE architectures. By introducing stochastic co-activation sampling and a hierarchical router loss, the authors aim to alleviate the mismatch between the training-time expert co-occurrence distribution and the inference-time routing patterns when increasing the number of activated experts. Experiments on several MoE backbones (LoRA-MoE, OLMoE, DeepSeek-V2-Lite) demonstrate smoother performance scaling and moderate improvements under larger inference budgets.

**Strengths:**

- The work systematically investigates the phenomenon that scaling up the number of activated experts at inference time causes performance degradation, which is empirically well presented.

- The proposed stochastic sampling mechanism and hierarchical router regularization are simple to implement and compatible with existing MoE frameworks.

- The method yields clear empirical gains with nearly no additional training cost, offering some practical relevance for deployment scenarios with dynamic compute budgets.

**Weaknesses:**

- The paper does not provide sufficient comparison against directly fine-tuning or training a model with a larger number of active experts 𝐾 under the same data and compute budget. Empirically, the reported improvements are modest (mostly 1–2 points) and statistically insignificant relative to directly increasing 𝐾. Since the training cost difference is marginal, simply fine-tuning with a larger  𝐾 remains a simpler and equally effective alternative, which undermines the necessity and originality of EMoE.

- The proposed stochastic co-activation sampling and hierarchical router loss offer only minor methodological variations over existing ideas such as DropMoE, expert dropout, or router regularization.

- The elasticity of EMoE is restricted: the method performs reasonably only within roughly 2–3× of the training (K_{\text{train}}); performance deteriorates beyond this range. The method is also sensitive to the hyperparameter (K_{\text{ideal}}), requiring task-specific tuning. Moreover, EMoE does not address fundamental MoE bottlenecks such as communication overhead or expert load imbalance, raising doubts about its scalability and practicality for large-scale deployment.

**Questions:**

- How does EMoE fundamentally differ from directly fine-tuning or training a model with a larger number of active experts?

- Can the authors provide quantitative evidence that stochastic co-activation sampling improves expert collaboration, rather than simply introducing noise? For example, can you measure diversity or mutual information across experts before and after applying EMoE?

---

> ### Author Response · Authors · 2025-11-18
> **Response to UsPv (1/3)**
>
> We sincerely thank reviewer UsPv for the thorough review of our work and for raising insightful questions. We greatly appreciate your recognition of our systematic study on the issue of performance degradation when increasing the value of k during inference, as well as the simplicity and practicality of our proposed method, EMoE. Below, we address each of your concerns in detail.
>
> ## W1.1 & Q1: Comparison with Models Trained Using Higher k Values
> ### (A) Results of Top-k Models Trained with Larger k
> We fully agree that comparing EMoE to models directly trained with higher k values is essential for assessing the value of our approach. Here, we compare our EMoE model (trained with k_train=2) against a standard Top-k model trained directly with a larger number of experts (k_train=6).
>
> | Model | k'=1 | k'=2 | k'=4 | k'=6 | Memory Cost (Per GPU) |
> | --- | --- | --- | --- | --- | --- |
> | Top-k (k_train=6) | 42.87 | 46.26 | 48.3 | 49.14 | 76.0GB |
> | EMoE (w/o $\mathcal{L}\_\text{HR}$, k_train=2) | 45.19 | 47.79 | 48.81 | 49.08 | 43.4GB |
> | EMoE (w/ $\mathcal{L}\_\text{HR}$, k_train=2) | 45.68 | 48.22 | 49.00 | 49.50 | 43.4GB |
>
> The experimental results clearly demonstrate the trade-offs. At its native inference budget (k′ = 6), the k_train = 6 Top-k model is only slightly better than EMoE without $\mathcal{L}\_\text{HR}$. However, it suffers from two important limitations:
>
> 1. **Lack of flexibility:** When **reverting to the conventional sparse MoE configuration (e.g., k′ = 2)** for lower-cost inference, its **performance drops sharply**, making it unsuitable for dynamically changing deployment environments.
> 2. **Higher training cost:** Increasing k_train from 2 to 6 leads to **approximately 3× FLOPs and more activation memory** **in the MoE layers** during training, which grows increasingly significant as model size scales.
>
> In contrast, **EMoE achieves reliable performance across a broad range of inference budgets while being trained at a lower cost**.
>
> ### (B) Scaling Up k_train in EMoE for Further Gains
>
> In Figure 8 on page 10 of our revised manuscript, we additionally evaluate EMoE with higher training budgets (e.g., k_train=3,4). These experiments show EMoE continues to improve in the large-k′ regime as k_train increases.
>
> This demonstrates that:
>
> 1. EMoE’s performance is **not limited** by training with a single k_train.
> 2. EMoE **benefits from more compute**, while standard Top-k **loses flexibility** as k_train grows.
> 3. EMoE thus provides a strictly more favorable trade-off between training cost and inference flexibility.
>
> **Taken together, these results indicate that EMoE maintains reliable performance not only when the inference budget k' exceeds the k_train, but also when it is reduced below it**, whereas standard Top-k models are inherently tied to the k_train.
> ### (C) Deployment requires one model that supports multiple inference budgets
>
> In real systems, a MoE model is often deployed across hundreds or thousands of GPUs. Retraining and redeploying a new model for each k′ setting is impractical. Standard Top-k models inherently bind training and inference to the same k, causing degradation whenever k′ differs.
>
> EMoE is designed precisely for this scenario: a single model can reliably cover a wide range of budget points (both lower and higher than the training k), **without the need for retraining or deploying multiple versions.** Thus, the problem EMoE solves is not just “how to fine-tune more cheaply,” but rather: **how to ensure that a single MoE model can support multi-budget inference conveniently.**
> ## W1.2: On the magnitude of performance improvements
>
> We acknowledge that the absolute improvements of EMoE are typically 1–3 points. However, (1) this magnitude is **in line with most prior MoE optimization work[1-4]**, where typical gains are also in the 1–3 point band; (2) **the key contribution of EMoE lies not in the size of the gains but in reversing the degradation trend** that standard Top-k MoE reliably *drops* when (k' > k_train), whereas EMoE turns this into **monotonic improvement**; and (3) this behavior is **consistently observed across 30+ configurations** spanning **7B–21B models (including the newly added 21B ERNIE-4.5 in Table 1 on page 7)**, four MoE architectures (LoRA-MoE and three MoE base models), nine benchmarks, and **multiple data scales (now including 100K and 200K settings in Table 6 on page 18)**, **without a single exception**. Such consistency indicates that EMoE addresses a structural failure mode in inference-time scaling rather than yielding marginal or dataset-specific improvements.
>
> [1] Zeng et al., AdaMoE: Token-Adaptive Routing with Null Experts for Mixture-of-Experts Language Models
>
> [2] Huang et al., Harder Tasks Need More Experts: Dynamic Routing in MoE Models
>
> [3] Qiu et al., Demons in the Detail: On Implementing Load Balancing Loss for Training
> Specialized Mixture-of-Expert Models
>
> [4] Dai et al., StableMoE: Stable Routing Strategy for Mixture of Experts

---

> ### Author Response · Authors · 2025-11-18
> **Response to UsPv (2/3)**
>
> ## W2: Comparison with Expert Dropout and Router Regularization
>
> We appreciate the reviewer’s suggestion to compare our approach with existing methods, which helps clarify the unique contributions of EMoE. Our method is fundamentally different from expert dropout or DropMoE, both in motivation and mechanism.
>
> - **Different Motivations:**
>     - Dropout-based methods are primarily motivated by **regularization under a fixed compute budget.** They aim to enhance model **robustness** by randomly dropping a small subset of components.
>     - In contrast, EMoE addresses a novel challenge: **how to enable elastic scalability of computation in MoE models at inference time**. Our goal is to train experts to collaborate effectively under changeable inference budgets rather than those used during training.
> - **Different Mechanisms (“Subtractive” vs. “Additive”):**
>     - Dropout-based methods perform a “**subtractive**” operation by dropping a few experts from the already selected top-k, focusing on making the model insensitive to the absence of the dropped experts.
>     - Our core mechanism, **Stochastic Co-activation Sampling**, is “**additive**”: we sample and train small k-size combinations from a candidate pool (k_ideal, typically more than twice k). This is designed to repeatedly expose the model to new combinations of experts likely to appear in higher-budget scenarios, thereby reducing the measured co-occurrence discrepancy as shown in our paper, which is fundamentally different from the logic of Dropout-style regularization.
>
> Building on this, both components of EMoE are tightly integrated to serve the central goal of elasticity. Similarly, the $\mathcal{L}\_\text{HR}$ design aims to enhance the router’s ranking consistency across different budgets, working together with stochastic co-activation sampling to achieve this objective.
>
> ## W3.1: Limited Range of Elasticity and Hyperparameter Sensitivity
>
> We acknowledge that the effective range of elasticity for EMoE is not unlimited, as discussed in Figure 5 on page 9 and the limitations section of our paper. Our main contribution is to **significantly extend the nearly inelastic range of standard MoE models to a practical 2-3x range**, which already marks an improvement and covers a wide range of real-world application scenarios.
>
> Regarding the sensitivity of the k_ideal hyperparameter, we would like to clarify that **all models in our experiments are trained on a unified mixed-instruction fine-tuning dataset and tested on various downstream tasks**, using a fixed k_ideal setting per model. **Only minor variations in k_ideal existed between models, not between tasks.** Our results (Figures 5 and 10) demonstrate that EMoE achieves stable and strong performance across a **fairly wide range** of k_ideal values (from 2x to 4x) for base models. No per-task adaptation were made, which attests to the robustness of our approach. In the revised manuscript, we have made the “suitable region” in Figures 5 and 10 more explicit to better illustrate this point.
>
> ## W3.2: Motivation Does Not Involve Communication Overhead or Load Balancing
>
> Your observation is correct. Our work does not aim to address communication overhead or load balancing issues in MoE. Instead, our research focuses on an **orthogonal direction:** how to enable **inference-time dynamic adjustment of computational density** on deployed models with fixed communication patterns. From a deployment perspective, our method allows a single model to serve a variety of budget requirements (e.g., high-precision offline processing with k'=8 and low-latency online interaction with k'=2). This represents a significant step forward in the practical usability of vanilla MoE models.

---

> ### Author Response · Authors · 2025-11-18
> **Response to UsPv (3/3)**
>
> ## Q2: Further Evidence for the Effectiveness of Stochastic Co-activation Sampling
>
> This is a good question. Following your suggestion, we conduct new quantitative analyses to demonstrate that stochastic co-activation sampling indeed improves expert collaboration and functional specialization across three domains: mathematics (GSM8K), commonsense (Hellaswag), and code (Humaneval). We compute the mutual information (MI) between expert selection and the task domain; a higher MI indicates stronger specialization. Specifically, we calculate the aggregated global expert usage probability P(e) and the domain-conditional probability P(e|d) (see Lines 969–977 in the revised paper for detailed calculations):
>
> | Model Setup | Mutual Information | $\Delta$ |
> | --- | --- | --- |
> | Baseline (Standard Top-k) | 0.0473 | - |
> | + Co-activation Sampling (w/o $\mathcal{L}\_\text{HR}$) | 0.0603 | **+27.5%** |
> | + Hierarchical Router Loss (Full EMoE) | 0.0630 | +4.5% |
>
> The results show that **stochastic co-activation sampling is the main driver of expert specialization, and $\mathcal{L}\_\text{HR}$ further refines and enhances this effect, leading to the highest degree of specialization**. This further confirms that $\mathcal{L}\_\text{HR}$ is not an isolated component but a synergistic, hierarchy-aware component crucial for true elasticity.
>
> Thank you once again for your valuable feedback. We are committed to refining our work based on your insights.
>
> If there are any remaining concerns or questions that we have not yet addressed, please do not hesitate to let us know. We are more than willing to provide any additional materials or clarifications that may be necessary.

---

> ### Author Response · Authors · 2025-11-26
>
> Dear Reviewer UsPv,
>
> Thank you again for your thoughtful and constructive reviews. We hope our responses have helped clarify the concerns raised, and we truly appreciate the opportunity to improve our work.
>
> If there are any remaining questions or suggestions, we would be grateful to hear them. Your feedback is incredibly valuable to us.
>
> ICLR 2026 Submission16399 Author

---

### Official Review · Reviewer_hP7x · 2025-11-04

**Soundness:** 3
**Presentation:** 4
**Contribution:** 3
**Rating:** 8
**Confidence:** 4

**Summary:**

Typically, MoE models use a fixed number of experts (k) during both training and inference. The authors identify that if one tries to activate more experts at inference (k′ > k), performance unexpectedly drops after a slight increase, rather than improving. They trace this to a lack of learned collaboration among experts beyond the combinations seen during training. The paper proposes (1) Stochastic co-activation sampling, which during training randomly samples diverse combinations of experts (from a larger candidate pool) to co-activate, thereby teaching experts to work together in various groupings. (2) Hierarchical router loss, a regularization term (based on KL divergence) that encourages the router’s output distribution to be sharp (non-uniform), producing a clear expert ranking for each token. EMoE is shown to extend the inference expert count to about 2–3× the training count effectively.

**Strengths:**

Well-jusified problem to develop a method that offers large-k flexibility at inference, while avoiding its training cost. A good motivation example.

The authors convincingly diagnose this as a result of experts never having learned to work together in those larger groupings.

During training, instead of always using the top-k experts, EMoE occasionally samples a small subset from a larger top-$k_{ideal}$ pool. It is a simple but innovative solution.

The paper is well-written, clearly explained.

The experiments are extensive (Moe using FFN and Lora, nine diverse tasks). EMoE-trained models exhibit monotonic improvement as the number of inference experts increases, eliminating the drop seen in standard MoEs.

**Weaknesses:**

The motivation of the hierarchical router loss in view of the entire paper that rather focuses on dealing with the problem of lack of collaboaration is a little weak.

One missing comparison is to a model trained with a higher k from the start. For instance, if we train a model with k_train = 4 (using 4 experts per token) and use 4 at inference, how does that compare to an EMoE model trained with k_train = 2 but using 4 at inference?

**Questions:**

How does the method behave beyong the scenario with more than 3x experts at inference?

Are there any disadvanatges of having a too sharp expert distribution? given the theme of the paper, could making the router too sharp hurt diversity?

---

> ### Author Response · Authors · 2025-11-18
> **Response to hP7x (1/2)**
>
> We sincerely thank reviewer hP7x for their insightful comments and positive evaluation of our work. Your recognition of our problem definition, causal analysis, and the simplicity of our solution is highly encouraging. We are also very grateful for your constructive suggestions, which have helped us further clarify the contributions of our paper. Below, we address each of your concerns in detail.
>
> ## W1 Motivation of $\mathcal{L}\_\text{HR}$
>
> We greatly appreciate your suggestion to more closely link the motivation for $\mathcal{L}\_\text{HR}$ with the central theme of “expert collaboration” in our paper. We have reinforced this connection in the revised manuscript. **The goal of EMoE is to deliver reliable performance across varying inference-time budgets while preserving the standard training cost of conventional k configurations**, which requires solving **two interrelated challenges**:
>
> 1. Teaching experts how to collaborate effectively in diverse group sizes and combinations, a capability fostered by our *Stochastic Co-activation Sampling*. This training endows experts with the necessary “collaborative skills” for functioning in variable inference budgets.
> 2. Teaching the router how to select the most favorable expert combinations under different computational budgets, precisely where the *Hierarchical Router Loss ($\mathcal{L}\_\text{HR}$)* plays a pivotal role.
>
> While stochastic co-activation equips experts with collaborative skills, the router needs to identify the most effective combinations, especially under small activation budgets. Without $\mathcal{L}\_\text{HR}$, this selection mechanism can be unstable. As shown in our ablation (Table 3 on page 9), the variant without $\mathcal{L}\_\text{HR}$ performs noticeably worse at low activation budgets (k′ = 1 and k′ = 2), revealing that **router instability becomes the bottleneck even when expert collaboration is available**. $\mathcal{L}\_\text{HR}$ is designed to directly address this issue.
>
> As shown in Figure 7 on page 10 of our revised paper, we compare the expert co-occurrence matrices during inference (**k’=2, denote low activation budgets**) for models with and without $\mathcal{L}\_\text{HR}$, across two specialized domains: mathematical reasoning (GSM8K) and commonsense reasoning (HellaSwag). This figure clearly demonstrates the impact of $\mathcal{L}\_\text{HR}$ on the router’s selection strategy:
>
> - The model without $\mathcal{L}\_\text{HR}$ (right panel) displays a relatively diffuse co-occurrence pattern. This indicates that, in the absence of the loss, the router can activate collaborative experts but lacks decisiveness and consistency in picking favorable pairs for the given task, resulting in uncertainty in its selection strategy.
> - In contrast, the model with $\mathcal{L}\_\text{HR}$ (left panel) exhibits a sparse, highly concentrated co-occurrence pattern dominated by a few pronounced “hotspots.” This evidences that it successfully guides the router to consistently select favorable expert combinations for specific tasks.
>
> This optimization of the selection strategy is also reflected in information entropy: the entropy of the co-occurrence matrix without $\mathcal{L}\_\text{HR}$ is 7.1, while with $\mathcal{L}\_\text{HR}$ it is reduced to 5.8 for the math domain and 8.6 vs 7.9 for the commonsense domain, indicating a **nearly 10%-20% decrease in uncertainty**. In summary, $\mathcal{L}\_\text{HR}$ is not an orthogonal add-on, but a **crucial component that stabilizes the router’s combinatorial selection under varying budgets**, particularly at low activation budgets, thereby enabling EMoE to achieve robust and truly elastic behavior.

---

> ### Author Response · Authors · 2025-11-18
> **Response to hP7x (2/2)**
>
> ## W2: Comparison to Top-k Models Trained with Larger k
>
> This is an excellent suggestion and a crucial baseline for our study. Following your advice, we compare our EMoE model (trained with k_train=2) against a standard Top-k model trained directly with a larger number of experts (k_train=6).
>
> | Model | k'=1 | k'=2 | k'=4 | k'=6 | Memory Cost (Per GPU) |
> | --- | --- | --- | --- | --- | --- |
> | Top-k (k_train=6) | 42.87 | 46.26 | 48.3 | 49.14 | 76.0GB |
> | EMoE (w/o $\mathcal{L}\_\text{HR}$, k_train=2) | 45.19 | 47.79 | 48.81 | 49.08 | 43.4GB |
> | EMoE (w/ $\mathcal{L}\_\text{HR}$, k_train=2) | 45.68 | 48.22 | 49.00 | 49.50 | 43.4GB |
>
> The experimental results clearly demonstrate the trade-offs. At its native inference budget (k′ = 6), the k_train = 6 Top-k model is only slightly better than EMoE without $\mathcal{L}\_\text{HR}$, and performs on par with the full EMoE model. However, it suffers from two important limitations:
>
> 1. **Lack of flexibility:** When **reverting to the conventional sparse MoE configuration (e.g., k′ = 2)** for lower-cost inference, its **performance drops sharply**, making it unsuitable for dynamically changing deployment environments.
> 2. **Higher training cost:** Increasing k_train from 2 to 6 leads to **approximately 3× FLOPs and more activation memory** in the MoE layers during training, which grows increasingly significant as model size scales.
>
> In contrast, **EMoE achieves reliable performance across a broad range of inference budgets while being trained at a lower training cost**. We additionally experiment with **increasing the training budget for EMoE (e.g., k_train = 3 and k_train = 4) in Figure 8 on page 10 of our revised paper, and observe further improvements in elasticity and large-k′ performance**. This confirms that EMoE is complementary to increasing k_train: it benefits from larger training budgets, but does not rely on them to achieve reliable multi-budget behavior.
>
> **Taken together, these results indicate that EMoE maintains reliable performance not only when the inference budget exceeds the training-time budget, but also when it is reduced below it,** **whereas standard Top-k models are inherently tied to the training-time budget**. In practical deployment, **only a single EMoE model needs to be deployed** to achieve reliable performance across various inference costs. In contrast, a conventionally trained Top-k MoE model can only be used under its specific computational budget. If different inference costs are required, multiple Top-k MoE models trained with different budgets would have to be deployed, which is highly impractical.
>
> ## Q1: Performance Beyond 3x the Number of Experts
>
> Thank you for this forward-looking question. Our analysis in Figure 5 on page 9 shows that EMoE achieves effective elasticity typically within 2-3 times the k value used during training. Beyond this range, performance plateaus. Our main contribution lies in expanding the effective elasticity range of the model from none to a practical 2–3x window, which covers a wide range of real-world scenarios requiring dynamic computation allocation.
>
> ## Q2: Impact on Diversity
>
> This is a very insightful question that addresses a core design consideration of $\mathcal{L}\_\text{HR}$. We specifically chose reverse KL divergence for $\mathcal{L}\_\text{HR}$ to avoid making the distribution overly sharp. To evaluate $\mathcal{L}\_\text{HR}$’s impact on expert diversity, we conduct an experiment measuring the router’s specialization across three domains: mathematics (GSM8K), commonsense (HellaSwag), and code (Humaneval). We computed the **Mutual Information (MI)** between expert selection and task domain; a higher MI indicates stronger specialization. Specifically, we calculate the aggregated global expert usage probability P(e) and the domain-conditional probability P(e|d):
>
> | Model Setup | Mutual Information | $\Delta$ |
> | --- | --- | --- |
> | Baseline (Standard Top-k) | 0.0473 | - |
> | + Co-activation Sampling (w/o $\mathcal{L}\_\text{HR}$) | 0.0603 | **+27.5%** |
> | + Hierarchical Router Loss (Full EMoE) | 0.0630 | +4.5% |
>
> The results show that **stochastic co-activation sampling is the main driver of expert specialization, and $\mathcal{L}\_\text{HR}$ further refines and enhances this effect**, leading to the highest degree of specialization. This further confirms that the loss is not an isolated component but a synergistic, hierarchy-aware component crucial for true elasticity.
>
> Thank you once again for your valuable feedback. We are committed to refining our work based on your insights.
>
> If there are any remaining concerns or questions that we have not yet addressed, please do not hesitate to let us know. We are more than willing to provide any additional materials or clarifications that may be necessary.

---

### Author Response · Authors · 2025-11-18
**General Response to All Reviewers (Summary of Paper Modifications)**

We sincerely thank all reviewers for the detailed, constructive, and highly valuable feedback. Your comments were exceptionally thorough and have significantly improved the quality of this work.

To address the key concerns, particularly the request for direct comparisons with larger k_train baselines, clearer motivation for the hierarchical routing loss, empirical evidence at larger data scales, and broader validation, we have made the following revisions:

## Experimental Additions

- **Added larger k_train baselines in Table 5 (Page 18).**

    We now include a direct comparison between the Top-k baseline (k_train=6) and EMoE (k_train=2).

    The results show that while the k_train=6 model performs comparably to EMoE at its native budget (k' = 6), it suffers a sharp degradation at lower budgets (k' = 2), which correspond to the conventional sparse activation mode in practical MoE systems, and it simultaneously incurs substantially higher training FLOPs and peak GPU memory.

- **Added EMoE (k_train) scaling results in Figure 8 (Page 10).**

    Using higher training budgets (k_train=3) and (k_train=4), we show that both peak performance and elasticity continually improve.

- **Added an analysis of the Hierarchical Routing Loss ($\mathcal{L}\_\text{HR}$) in Figure 7 (Page 10).**

    We visualize its effect on expert co-activation matrices and quantify the entropy changes on GSM8K and HellaSwag.

    These results demonstrate that $\mathcal{L}\_\text{HR}$ significantly improves routing consistency under low-budget inference.

- **Added experiments on a 21B-parameter MoE model (ERNIE-4.5-21B-A3B) in Table 1 (Page 7).**

    The results, included in Table 1, confirm that EMoE remains effective at much larger parameter scales, demonstrating strong scaling with model size.

- **Added experiments using larger instruction-tuning datasets (100K and 200K) in Table 6 (Page 18).**

    We confirm that the elasticity window remains stable even when the data scale is increased by 4×, indicating that EMoE is applicable across different dataset sizes, and, notably, elasticity can be achieved with 50K samples, demonstrating the plug-and-play nature of the method and its low overhead.

- **Added mutual-information analyses across mathematical, commonsense, and coding tasks (GSM8K / HellaSwag / HumanEval) in Table 7 (Page 19).**

    These results show that both EMoE sub-methods consistently improve expert specialization across domains.


## Textual Revisions

- Revised **Figure 1 and its explanation** to more clearly highlight the performance collapse of large k_train baselines under low inference budgets (Introduction, Sec 3.2).
- Clarified the **motivation for the hierarchical routing loss**, emphasizing that it complements stochastic co-activation sampling and addresses inaccurate routing under low-budget inference.
- Revised **Figures 5 and 10** to more clearly illustrate the flexibility of EMoE across a wide range of hyperparameter configurations.

Thank you once again for your insightful comments.

---

### Author Response · Authors · 2025-12-01
**Summary of Rebuttal**

We summarize how each reviewer’s concerns (W/Q) are addressed. We added new experiments/analyses for larger k_train baselines, elasticity range, $\mathcal{L}\_\text{HR}$’s role, data/model scaling, and co-activation effects (Table 1/5/6/7, Figure 7/8). **Reviewer kgGr explicitly acknowledged these additions and increased the rating to 6 on November 20, while we have not yet received responses from the other reviewers.**
| Reviewer | Concern | Response |
| --- | --- | --- |
| hP7x | Role of $\mathcal{L}\_\text{HR}$ (W1) | Clarified the hierarchical routing loss ($\mathcal{L}\_\text{HR}$) is for stabilizing routing decisions across different inference budgets. Fig. 7 and ablations (Tab. 3) show that $\mathcal{L}\_\text{HR}$ substantially improves performance at low budgets (k' = 1,2) and enhances routing consistency. |
| hP7x | Larger k_train baseline (W2) | Added larger-k_train baselines in Tab. 5, comparing a Top-k model with k_train=6 to EMoE with k_train=2 under multiple inference budgets. EMoE matches the large-k model at its native budget (k'=6) but significantly outperforms it at low budgets (k'=2) while using lower training compute and memory. Moreover, Fig. 8 shows that increasing EMoE’s training budget (e.g., k_train = 2 → 3 → 4) continues to improve both peak performance and elasticity, indicating that our method is not restricted to k_train settings. |
| hP7x | Beyond 2–3× elasticity (Q1) | Revised Fig. 5: EMoE provides a practical ~2–3× k_train elasticity window; beyond that, gains saturate. |
| hP7x | Diversity under $\mathcal{L}\_\text{HR}$ (Q2) | We analyzed mutual-information metrics (Tab. 7) and showed that EMoE increases expert specialization. |
| UsPv | Larger k_train baseline (W1/Q1) | This concern is addressed by the same larger-k_train experiments as in hP7x (W2). |
| UsPv | Relation to dropout-based method (W2) | Clarified goal/mechanism: dropout-based methods regularize for fixed budgets; EMoE targets inference-time elasticity. Dropout is subtractive (drop a few experts while ensuring that most experts remain activated and trained); EMoE is additive (sample and train $k$-expert combinations from a much larger $k_{\text{ideal}}$) + $\mathcal{L}\_\text{HR}$ to stabilize routing across budgets. |
| UsPv | Elasticity range & $k_{\text{ideal}}$ (W3) | Revised Fig. 5/10: stable window gains across a broad range of $k_{\text{ideal}}$. In addition, we clarified that EMoE is explicitly targeting a different, practically important dimension of MoE deployment: inference-time budget elasticity under a fixed communication topology, where a single MoE model can be used for different budgets. In this sense, EMoE is novel and complementary to existing work on communication and load-balancing optimizations. |
| UsPv | Co-activation effectiveness (Q2) | Added MI on GSM8K/HellaSwag/HumanEval (Tab. 7): co-activation increases specialization; co-activation + $\mathcal{L}\_\text{HR}$ further increases MI (stronger expert–domain alignment). |
| kgGr | Larger k_train baseline (W1/W2/Q1) | This concern is addressed by the same larger-k_train experiments as in hP7x (W2). |
| kgGr | Scaling with model size & k_train (Q2) | Added Fig. 8: increasing EMoE k_train (3,4) yields monotonic improvements (peak + elasticity). Added experiments in Tab. 1: benefits persist on a larger 21B MoE model. |
| rgdX | Small SFT data scale (W1) | Added 100K/200K instruction-tuning (Tab. 6): elasticity remains stable with larger data, and importantly, we show that strong elasticity can already be achieved with 50K samples, highlighting EMoE’s plug-and-play, low-overhead nature. |
| rgdX | Larger k_train baseline (W2/Q2) | Added large-k_train Top-k baselines (i.e., k_train=6 LoRA-MoE; k_train=16 OLMoE). They underperform at conventional sparse budgets and cost more; EMoE stays strong across low/high k', and only one model needs to be deployed. |
| rgdX | Role of $\mathcal{L}\_\text{HR}$ (W3) | This concern is addressed by the same revision as in hP7x (W1). |
| rgdX | Pretraining-stage integration (Q1) | We clarified our design choice: EMoE is intentionally designed as a post-hoc SFT / continued-tuning mechanism that can be applied to existing pretrained MoE models without modifying the expensive pretraining phase. We also show effectiveness across both LoRA-MoE and FFN-MoE downstream setups, suggesting that EMoE is broadly applicable as a lightweight and plug-and-play adaptation method. |

Overall, EMoE is a lightweight, plug-and-play adaptation for pretrained MoE models without modifying the pretraining pipeline. It enables inference-time elasticity, allowing a single model to perform well under a wide range of budgets k' (roughly a 2–3× window over k_train). Moreover, EMoE shows clear scaling behavior: as we increase training budgets or model size, both peak performance and elasticity consistently improve. This combination of practical deployability, compute efficiency, and scalable gains distinguishes EMoE from prior approaches.

---

### Meta-Review · Area_Chair_Jcn7 · 2025-12-12

**Summary:**

The manuscript is reviewed by four reviewers, of whom three rated below the acceptance threshold, in which one raised the rate after rebuttal. On the other hand, one reviewer recommended acceptance.
The reviewers pointed out some weaknesses, such as lack of novelty compared to existing MOE methods and experimental evaluations on different $k$ values. Moreover, there was some ambiguity in the presentation, such as the hierarchical router loss and experimental setup.

After carefully considering the rebuttal and the revised paper. I noticed the authors provided a conceptual distinction between EMoE and existing dropout-based methods. However, the reviewer concern about methodological similarity remains unresolved. Indeed, the stochastic co-activation sampling can be interpreted as a structured variant of existing expert-dropout techniques, and the hierarchical router loss resembles previously studied router regularization mechanisms.
Additionally, the rebuttal does not provide empirical comparisons with DropMoE/expert-dropout nor with router regularization baselines, nor does it provide a formal analysis demonstrating methodological novelty. Therefore, the paper contribution appears incremental compared to existing MoEs.

**Reviewer Concerns:**

The concerns of hP7x and kgGr are mostly addressed. However, UsPv concerns about the novelty and the loss function were not addressed truly. For the reviewer rgdX comments, I am satisfied with most of the authors' responses. However, the author's response was not convincing regarding model performance after tuning. So, still there is a question of whether EMoE provides benefit over simply fine-tuning a standard MoE at the desired inference budget.

**Reviewer Scores:**

hP7x and kgGr concerns are addressed. But, for the other two reviewers, there are some points that remain.

---

### Decision · Program_Chairs · 2026-01-26

Reject